



# The De-Icing Comparison Experiment (D-ICE): A study of broadband radiometric measurements under icing conditions in the Arctic

Christopher J. Cox[1], Sara M. Morris[1], Taneil Uttal[1], Ross Burgener[2], Emiel Hall[2,3,4], Mark Kutchenreiter[5],
Allison McComiskey[6], Charles N. Long[2,3,4,*], Bryan D. Thomas[2], and James Wendell[2]

[1]NOAA Physical Sciences Laboratory (PSL), Boulder, Colorado, 80305, USA
[2]NOAA Global Monitoring Laboratory (GML), Boulder, Colorado, 80305, USA
[3]Cooperative Institute for Research in Environmental Sciences (CIRES), Boulder, Colorado, 80305, USA
[4]University of Colorado, Boulder, Colorado, 80305, USA
[5]National Renewable Energy Laboratory (NREL), Golden, Colorado, 80401, USA
[6]Brookhaven National Laboratory (BNL), Upton, New York, 11973, USA
[*]retired, deceased

Correspondence: Christopher J. Cox (christopher.j.cox@noaa.gov)

**Abstract.** Surface-based measurements of broadband shortwave (solar) and longwave (infrared) radiative fluxes using thermopile radiometers are made regularly around the globe for scientific and operational environmental monitoring. The occurrence of ice on sensor windows in cold environments – whether snow, rime, or frost – is a common problem that is difficult to prevent as well as difficult to correct in post-processing. The Baseline Surface Radiation Network (BSRN) community recognizes radiometer icing as a major outstanding measurement uncertainty. Towards constraining this uncertainty, the De-Icing Comparison Experiment (D-ICE) was carried out at the NOAA Atmospheric Baseline Observatory in Utqiaġvik (formerly Barrow), Alaska, from August 2017 to July 2018. The purpose of D-ICE was to evaluate existing ventilation and heating technologies developed to mitigate radiometer icing. D-ICE consisted of 20 pyranometers and 5 pyrgeometers operating in various ventilator housings alongside operational systems that are part of NOAA's Barrow BSRN station and the U.S. Dept. of Energy Atmospheric Radiation Measurement (ARM) Program North Slope of Alaska and Oliktok Point observatories. To detect icing, radiometers were monitored continuously using cameras, with a total of more than 1 million images of radiometer domes archived. Ventilator and ventilator/heater performance overall was skilful with the average of the systems mitigating 77% of icing and many being 90+% effective. Ventilators without heating elements were also effective and capable of providing heat through roughly equal contributions of waste energy from the ventilator fan and adiabatic heating downstream of the fan. This provided ~0.6 C of warming, enough to subsaturate the air up to a relative humidity (w.r.t. ice) of ~105%. Because the mitigation technologies performed well, a near complete record of verified ice-free radiometric fluxes were assembled for the duration of the campaign. This well-characterized data set is suitable for model evaluation, in particular for the Year of Polar Prediction (YOPP) first Special Observing Period (SOP1). We used the data set to calculate short- and long-term biases in iced sensors, finding that biases can be up to +60 Wm² (longwave) and -211 to +188



Wm² (shortwave). However, because of the frequency of icing, mitigation of ice by ventilators, cloud conditions, and the
timing of icing relative to available sunlight, the biases in the monthly means were generally less than the aggregate uncertainty
attributed to other conventional sources.

## 1. Introduction

Radiative fluxes are fundamental environmental observations made regularly from the earth's surface using
thermopile radiometers. In cold climates, ice from vapor deposition (frost), contact freezing of supercooled droplets (rime) and
accumulation of snow are all commonly observed by station personnel to obscure sensors and manual cleaning of sensor domes
is a routine activity. Icing is the source of one of the least constrained, outstanding uncertainties in broadband radiometry in
cold climates. For radiometers mounted facing upwards, ice generally increases the measured longwave downwelling (LWD)
flux because the brightness temperature of the contaminating ice is typically larger than that of the sky. The relatively cold
background of the sky also facilitates radiative cooling of the sensor window, which exacerbates icing relative to instruments
pointed towards the ground. Biases can be both negative or positive in affected shortwave downwelling (SWD) fluxes by
attenuation or scattering of incident light, respectively. The magnitude of the instantaneous errors has been reported to be up
to 80 W m$^{-2}$ in LWD (Persson et al. 2018) and more than 100 W m$^{-2}$ in SWD (van den Broke et al. 2004; Matsui et al. 2013).
Despite these large biases, post-processing of data is hampered by the fact that the signal from data contaminated by ice is
difficult to distinguish from the signal caused by clouds. This is particularly problematic for LWD (Lanconelli et al. 2011).
Since icing occurs under specific meteorological conditions, even if affected data are successfully removed, the lost data
constitutes a climatological bias in the record. Therefore, the standard procedure of daily cleaning (McArthur 2005) is
insufficient and it is sometimes necessary to fill data gaps caused by icing with surrogate values (e.g., Persson et al. 2018). It
is desirable to identify a solution that prevents the formation of ice in the first place as well as to quantify the biases in
contaminated measurements to advance interpretation of data already collected.

     Recognition of the problem and mitigation attempts have been reported since the earliest era of polar radiometric
observations more than five decades ago (e.g., Koerner et al. 1963). Since then, engineering solutions have been pursued by
research institutes and industry, largely independently and in parallel. In practice, because the nature of the measurement is
sensitive to thermal instabilities within the instruments (e.g., Michalsky et al. 2017), the application of heat as an ice-mitigation
technique has limitations. While progress has been made, to this day there is still no agreed-upon approach. The needs of the
scientific community also increasingly require high-quality measurements from stations capable of being autonomous for
weeks or months at a time. Thus, an automated, low-power solution to the icing problem is sorely needed.

     The Baseline Surface Radiation Network (BSRN) (Ohmura et al. 1998; Driemel et al. 2018), under the auspices of
the World Meteorological Organization (WMO), is a global network for surface-based radiometric observations; the BSRN is
traceable to the world calibration standard, managed using commonly adopted practices and is strategically distributed for
global coverage. There are six current and former BSRN stations in the Arctic, three in Antarctica and numerous stations at



lower latitudes that are located at high elevations and/or experience icing conditions seasonally. In 2008, BSRN established the Cold Climates Issues Working Group (CCIWG) to address uncertainties in cold regions, including icing (Lanconelli et al.

2011). Several BSRN stations affected by icing have reported increased data capture rates using ventilators, including the Sonnblick station in the Austrian Alps (Weisser 2016) and the Georg von Neumayer station in Antarctica (BSRN 2016). The U.S. Department of Energy (DoE) Atmospheric Radiation Measurement (ARM) Program North Slope of Alaska Radiometer Campaign also reported that high-flow ventilation was a useful technique, but that ice mitigation was further improved when the air was also heated (BSRN 2012). A consensus in BSRN thus emerged that heating and ventilation are capable of mitigating

ice, but the effectiveness and uncertainties remained poorly quantified and the range of experiences reported by BSRN users indicated that more work was needed to constrain the attributes of effective designs (BSRN 2016).

To address these objectives, the NOAA Physical Sciences Laboratory (PSL) in partnership with the BSRN-CCIWG and NOAA Global Monitoring Laboratory (GML) carried out the De-Icing Comparison Experiment (D-ICE) to collect data suitable for assessing the influence of icing on the measurements and evaluating the status of ice-mitigation technology. D-

ICE was deployed at the GML Barrow Atmospheric Baseline Observatory near Utqiaġvik, Alaska, from August 2017 through June 2018. This location was chosen because a variety of icing conditions characteristic of high latitudes regularly occur there and it is home to two long-term operational stations, one BSRN (NOAA-GML) the other DoE-ARM. D-ICE collected new data at the NOAA observatory using a variety of radiometers and housings that have been developed to mitigate the formation of ice or are used in icing environments. The systems were contributed by academic and government research institutions as

well as development departments of commercial radiometer vendors and were installed alongside the existing operational suites. The systems were monitored continuously using cameras for the duration of the campaign.

In this manuscript, we describe D-ICE and associated data sets, which are available for future analyses. These data sets include a 10-15 minute resolution classification of the icing status of instruments, quality-controlled versions of the radiometric data with occurrences of icing retained and rejected (Cox, 2020a), and a verified ice-free "best-estimate" baseline

(Cox, 2020b) for comparison produced by the aggregate of the quality-controlled data. We use these data sets to analyse instantaneous and time-averaged biases caused by ice, to calculate ice-mitigation performance statistics for the participating systems, to discern some of the reasons for successful ice mitigation, and to gather insight for interpretation of ice-contaminated data.

## 95 2. Campaign

### 2.1. Experimental Design

D-ICE solicited contributions of radiometric ice-mitigation systems developed by research institutions and industry

manufacturers to be part of the campaign. In total, 26 systems were included, 21 housing pyranometers (measuring "global" SWD) and 5 housing pyrgeometers (measuring LWD). In addition to the Utqiaġvik BSRN station, D-ICE collaborated with



the two other operational stations; the DoE-ARM North Slope of Alaska (NSA) site located at Utqiaġvik ~150 m west of the BSRN station, and the DoE-ARM Oliktok Point (OLI) site located 250 km to the east, a partner campaign termed the D-ICE ARM Component (DICEXACO, Cox et al. 2019). The operational stations include global unshaded SWD, shaded LWD, and

diffuse (DIF) and direct (DIR) shortwave components using a shaded pyranometer and a pyrheliometer, respectively, mounted to a solar tracker. The focus of D-ICE was on upward-facing pyrgeometers and pyranometers. Pyrheliometer performance is not considered here, but a separate analysis was completed by DoE-ARM (Steufer et al. 2019).

The ice-mitigation strategy used by all contributors was some combination of heating and ventilation, in some cases supplied by separate housings in which radiometers were set and in others integrated into the instruments themselves. This

consistency in approach is not surprising. Though other methods have been proposed, such as automated alcohol rinses (e.g., Persson and Semmer 2010), the use of ventilators for controlling ice is pragmatic because ventilation is already regularly used for maintaining thermal homogeneity in the instrument. However, no specific criteria were given to potential contributors and D-ICE set up each system as instructed. Several sets of redundant housings were used with different radiometers or with only small modifications (see S1). All systems were powered using 12 or 24 VDC except for one 48 VAC heater. All fans were

powered by DC, which is less prone to propagation of added uncertainty into the signal (Michalsky et al. 2017), in particular from infrared loss in pyranometers (Dutton et al. 2001).

The instruments were installed on the east end of the GML observatory roof in a single line along a 4.9 m table positioned perpendicular to the predominant wind direction (Figure 1a) near the BSRN tracker. The table was constructed from aluminium with a top consisting of fiberglass resin to electrically isolate the systems. The BSRN global pyranometer was

positioned on this table. Refer to S1 for a complete record of system specifications and S2 for a list of modifications made during the course of the campaign. Individual radiometers are referenced in the text by their serial number and the ventilators by their model number. The positions of the systems are displayed in Figure 1b, labelled with numbers that are referenced where appropriate and cross-referenced in S1.

D-ICE data were collected using four Campbell Scientific CR1000 data loggers in individual logger boxes; most

systems were analog, but data was also logged digitally from seven sensors. Fan speeds and heating current were logged whenever possible. All data were recorded as one-minute averages of 1 Hz sampling except for the digital systems, which were switched to 0.5 Hz sampling on 26 October 2017 because lags that occurred in digital communications at temperatures below -10 C caused occasional missed scans. One-minute averages of wind speed and direction (Lufft 2d sonic), temperature (RTD) and relative humidity (Vaisala HMP155) were also recorded at the north end of the table (Figure 1b) to document

localized meteorological conditions complementary to that recorded routinely by NOAA-GML from a nearby tower.

Before deployment in June 2017, the radiometers were calibrated at the NOAA-GML calibration facility in Boulder, Colorado. Per standard procedures, the calibration data were collected without use of the ventilators, but did use the same data acquisition system that was later deployed. The digital systems were also included in this procedure for comparison, but were not assigned new calibration coefficients because it is impractical to do so. The pre- and post-campaign calibrations (S3) were



found to be within uncertainty for all instruments. The pre-campaign calibration values determined by NOAA-GML are used in the processing of the final data set.

All systems on the D-ICE table were monitored using three 720p low-light (0.1 lux) cameras in heated enclosures. The cameras recorded images every 15 minutes and were set up such that each captured approximately one third of the table. They were installed facing west (away from the predominant wind direction). Two 18 W LED flood lights were fixed to poles

to illuminate the table for the cameras. The lights were automatic and only on during low-light conditions. The cameras were functional and unobscured by ice for 97.6% of the campaign. ARM also installed cameras facing the trackers at OLI and NSA with 10 min sampling.

The BSRN and ARM operational systems received their routine daily maintenance procedures. Daily cleaning is performed to remove contaminants such as dust and salt residues, but also ice. Since one of the objectives of D-ICE was to

monitor icing it was important to allow icing events to unfold naturally. Therefore, the D-ICE radiometers were cleaned daily only when there was no ice present. Infrequently, in cases when ice persisted on a particular radiometer long after the end of an event, the ice was removed and recorded in the logbook. Interestingly, we found that icing can be induced by the very maintenance procedures that are designed to remove it. The use of alcohol (such as ethanol) to clean the domes is common practice and was documented during tests at D-ICE to sometimes result in immediate re-icing of the dome. The precise reasons

for this are not known, but it is likely a combination of refreezing meltwater from the ice that is residual, being slower to evaporate than the alcohol, and/or atmospheric vapor deposition induced by cooling of the dome from the evaporative process. Complete drying of the dome after cleaning was found to reduce this problem.

**2.2. Icing conditions during D-ICE**


During August and September, the temperatures were persistently above freezing with occasional light snow and frequent rain (Figure 1c). Significant icing was not observed until a prolonged cold period after 22 October, with only brief frosts prior on 28 September and 10 October. Warm temperatures and rain returned during the first week of November and more winter-like conditions prevailed only in the second half of the month. Autumn 2017 experienced record late freezing of

the Beaufort and Chukchi Seas (Overland and Wang 2018), with freezing beginning in earnest north of Utqiaġvik in late November. Because of the predominant onshore flow at Utqiaġvik, autumn temperatures there remain near-freezing until after the sea ice isolates the supply of heat from the ocean (Wendler et al. 2014, Cox et al. 2017) and the onset of snowpack is subsequently delayed in late freeze-up years (Cox et al. 2017). These conditions may have also contributed to the delay in the start of the 2017 icing season.

During the winter, both rime (usually from freezing fog) and frost were regularly observed (distinguished qualitatively from the images), spanning at total of 28.8 and 66.3 days, respectively. Frost events were more common, being identified 108 times compared to 11 rime events, but the duration of individual rime events was longer. The mean duration of frost events



was 0.61 (± 0.69) days and the mean duration of rime events was 2.6 (± 2.2) days. Diurnal (morning) frosts were commonly observed during spring.

During the campaign, 34.9% of the time that rime or frost was observed to be present in the vicinity of the D-ICE systems, the station meteorology indicated that the relative humidity with respect to ice (RHI) was < 100%. Note that this calculation is sensitive to the determination of the "end" of an icing event, which in practice was found subjectively using the camera images, and is therefore uncertain (refer to Section 2.3.4). Nevertheless, this implies that the sublimation period of icing events was approximately 1/3 of the duration of presence of ice. This period was longer for frost (41.6%) than rime (14.1%), which is surprising because riming events were generally observed with a thicker coating of ice than frost, but may be explained by the fact that rime events were much more persistent than frost. Conversely, 15% of the time during which no icing was observed the RHI exceeded 100%. Thus, RHI alone was not a reliable proxy for the presence or absence of icing.

**2.3. Data Processing**

Here we describe the processing of the data streams, beginning with review and classification of the images in 2.3.1 and then the radiometric data in 2.3.2, summarized in Table 1. The processed data streams were then used to produce a best-estimate ("BE") data set that is the average of the calibrated, bias-corrected, ice-free and quality controlled data streams in 2.2.3, from which uncertainties are derived in 2.3.4. A second iced-estimate data set was also made that received all of the same treatment except that occurrences of icing were retained for analysis.

**2.3.1. Processing of the D-ICE images**

The images captured approximately 780,000 views of the D-ICE radiometer domes with an additional 143,000 and 125,000 views captured by ARM at NSA and OLI, respectively. Images were captured of the BSRN global pyranometer and all 25 D-ICE radiometers, but not the instruments mounted on the BSRN tracker. At NSA and OLI, images of the global SWD, DIF and LWD tracker radiometers were captured, but only limited images of the pyrheliometers were made (see Steufer et al. 2019). The status of each dome in each image was recorded in a spreadsheet after manual review. Because of the large volume of images, this was done in movie form in one-month intervals, one radiometer at a time. The radiometer domes were classified as being wet (e.g., raindrops or melted ice and slush); containing frost, rime or snow accumulation; having accumulation of snow around the domes (but not on the domes); being wet with ethanol (used for cleaning); and (rarely) "other" contaminants, such as resting birds. Occurrences of rime and frost always took precedence in the classification. For example, in cases when snow and rime simultaneously affected a radiometer, the status of the instrument was recorded as rimed. Note that because the domes are hemispheric, the cameras were blind to some parts of the domes, though this was somewhat alleviated by the fact that the pyranometer domes are transparent and the pyrgeometer domes are relatively small and/or flat. All visible ice



regardless of amount or coverage was recorded. Thus, the classification was conservative; a snowflake or thick coating of rime were both flagged as iced. Camera downtime was also indicated.

To increase the robustness of the icing determinations, additional instances of ice were identified by comparing each of the data streams to the average of all the data streams and reviewing the images where anomalies were found. While this procedure successfully identified instances of icing that had been missed, the number of identifications increased by < 0.5%. This indicates that the original classification was sufficient to identify the icing that impacted data quality. However, the statistics compiled for the presence of ice include occurrences that were too minor to bias the measured signal and these occurrences were also both common and their identification subject to qualitative interpretation. The relevance of this limitation is discussed further in Section 4.1 where the statistics are reported.

### 2.3.2. Baseline data from the trackers

While the BSRN instruments that were mounted on the solar tracker were not imaged by the cameras, the tracker instruments provide important information for two reasons: First, the pyrgeometers were shaded, which reduces solar heating of the domes (Alados-Arboledas et al. 1988) and the magnitude of associated corrections that apply to some pyrgeometers (Albrecht and Cox 1977); and second, because SWD is more accurately represented by the sum (hereafter, "SUM") of the DIF and DIR due to increased calibration uncertainty in pyranometers from the direct beam at low sun angles (Michalsky et al. 1995). All BSRN data were quality controlled with manual screening and application of the relevant definitive tests described by Long and Shi (2008). The manual screening removed suspect data and shadows from station structures. The BSRN tracker measurements were supplemented where there was missing data by the SUM from the ARM QCRAD value-added product from the neighbouring ARM station, which is also based on Long and Shi (2008). The resulting data set was used as an intermediary processing step for two purposes: First, to provide a baseline to aid in identification of shadows on the D-ICE instruments and second to provide a statistical baseline for correcting or validating the aforementioned sources of uncertainty in the D-ICE measurements.

### 2.3.2. Quality controlling D-ICE data

#### 2.3.2.1. Light pollution

The amount of light pollution from the camera LEDs measured by the pyranometers was determined empirically for each instrument by comparison to night time periods on 19-20, 30 September and 1-2 October 2017 when the illumination was switched off. The calculated biases were then subtracted off in post-processing when the lights were on. These biases were small, ranging from ~0-1.5 $Wm^{-2}$ (mean, 0.35 $Wm^{-2}$).

#### 2.3.2.2. Shadows



Light poles, as well as some additional station structures, such as nearby aerosol inlet pipes were minimal obstructions to the view of the sky by the radiometers except for episodic appearances of shadows on clear days that reduced the signal in the pyranometers. The shadows occurred at different solar azimuth and zenith angles for each pyranometer and were only present when the sun was unobstructed by clouds. The times when each instrument was shadowed were identified by a reduction of normalized total irradiance signal exceeding -3% in the pyranometers relative to the SUM (Section 2.3.2) when

the direct beam accounted for at least 25% of the irradiance, which was qualitatively determined to be a suitable threshold for when shadows were observed to appear. This method for detection of shadows was supplemented with manual screening. Cases of shadows and ice co-occurring were treated as shadows and removed from the analysis of biases associated with ice. Approximately 1-4% of the data was removed, depending on the location of the instrument.

### 2.3.2.3. Outlier detection

Data from each D-ICE radiometer was processed with the same Long and Shi (2008) procedures as the operational systems. This was followed by manual screening. One radiometer (a1571), which is typically operated unventilated, was experimentally set in a ventilator that was later found to shadow its thermopiles. Another (26214) was out of level, but without the possibility of re-levelling after installation. Both of these radiometers are excluded from the radiometric analyses and BE

product, but are included in the analysis of the ventilator de-icing performance in Section 4.1.

The Long and Shi (2008) "QCRAD" approach is designed to identify outliers relative to the data stream being screened. It therefore relies on the assumption that most of the data falls within normal limits and is only sensitive to data that does not. Figure 2 shows examples of the "climatological configurable limits" for a D-ICE pyranometer in panel (a) and a pyrgeometer in panel (b). Since occurrences of icing rarely produce signals that fall outside the statistical distribution, the

255 spurious data are not readily captured by outlier-detection methodologies, such as QCRAD. For example, the 2nd level threshold for the definitive configurable QCRAD limit flagged < 3% of the iced data in Figure 2a (green line). Similarly, > 99% of the ~900 hours of iced LWD in Figure 2b passed the QCRAD climatologically configurable limit test.

### 2.3.2.4. Icing

The results from the processing of the D-ICE images were used to flag data contaminated by the presence of ice on the radiometer domes. This data was removed from the BE data set, but a second version with occurrences of icing preserved was needed to calculate the biases caused by the ice. To construct such a data set, only data that had been rejected for failing physically-possible limit tests or having been determined to be shadowed was removed while outliers flagged using other tests that were within physically-possible limits when ice was present were retained.

### 2.3.2.4. Infrared loss corrections

Infrared loss corrections were applied to pyranometers that exhibited night time offsets following the method of Dutton et al. (2001) (see Table 1), though the offsets observed during D-ICE were consistently small (generally < 3 Wm$^{-2}$).


Interestingly, two systems that feature air intake tubes extending below the ventilator fan (MeteoSwiss and Eigenbrodt 480) were found to have night time offsets in the CM11s they housed that were uncorrelated with the net longwave ($r^2 < 0.05$) but highly (negatively) correlated with wind velocity ($r^2 = 0.54$-$0.55$); this was also somewhat true for the CM11 and PSP in the PMOD ventilators ($r^2 = 0.22$ & $0.36$). What each of these systems have in common is that they were both heated and vulnerable to airflow obstructions. We interpret the source of this bias to be clogging from blowing snow that reduced aspiration thereby allowing the heating element to differentially warm the ventilator and radiometer case, which would be expected to produce a similar voltage offset as infrared loss. Clogging has also been reported in similar ventilators by the Sonnblick BSRN station and a longer set of inlet tubes were constructed there that alleviated the problem (Weisser 2016). For D-ICE, we corrected the affected pyranometers in the MeteoSwiss and Eigenbrodt 480 analogously to Dutton et al. (2001) using the wind measurements despite a limited understanding of how the bias would translate under sunlit conditions, but note that the corrections were small (the mean of the more severely-affected instrument was ~3 $Wm^{-2}$) and served to bring the measurements closer to the mean of the others.

### 2.3.3. Best Estimate Fluxes

The BE data set was produced by averaging the calibrated, bias-corrected, ice-free and quality controlled D-ICE data streams. For LWD, this consisted of all 8 upward-facing pyrgeometers (5 from D-ICE, 2 from NSA and 1 from BSRN). For SWD, this consisted of 17 pyranometers from D-ICE, the global BSRN and the global NSA. Since the sensitivity of a thermopile is not precisely isotropic, the calibration of global pyranometers is designed to be well-suited for the daily average but prone to varying errors through the day as the incident angle of the direct beam changes. Thus, to produce a BE, the average of the global pyranometers could be used to constrain the SUM from the trackers or the tracker measurements could be used to bias-correct the average of the global pyranometers. We chose to do the latter because the large number of included data streams produces a data set less prone to discontinuities and noise, and importantly is also verifiably ice-free. Thus, the SWD average was bias-corrected as a function of solar zenith angle (SZA) and the diffuse fraction using the SUM; the magnitude of the correction varied between 0.5% and 3% depending on the diffuse partitioning.

### 2.3.4. Uncertainty calculations

Uncertainty is estimated empirically as the 1σ spread of the one-minute average measurements where fluxes are available from at least two radiometers. Only the data that passed quality control procedures and was determined to be ice-free is included. Thus, the number of instruments and combinations of instruments included in the calculation varies minute to minute. However, no relationship between the number of radiometers included in estimated uncertainty is observable for either LWD or SWD. Nevertheless, the uncertainty may be best understood by its bulk properties: Uncertainty in LWD is shown in Figure 3a plotted against the mean net longwave flux from the pyrgeometer thermopiles (this differs from, but is correlated



with, the net longwave of the natural surface). The solid line is the mean of the 1 min resolution uncertainties in bins of 1 Wm$^{-2}$ of the net longwave and the shading is ±1σ of the same. Large negative net longwave values are indicative of clear skies

while the value is generally greater than -20 Wm$^{-2}$ in the presence of optically-thick clouds. The BSRN target uncertainty is 3 Wm$^{-2}$ (McArthur 2005); the average for D-ICE is 2.6 Wm$^{-2}$. The absolute accuracy of pyrgeometers at Utqiaġvik based on *in situ* calibration to a common standard has been shown previously to be ±2 Wm$^{-2}$ (Marty et al. 2003). The uncertainty is larger for clear skies and smaller for cloudy skies. This is not surprising because clear skies are anisotropic and spectrally-complex in the infrared. In the Arctic, the precipitable water vapor is commonly below 1 cm during clear skies. When this occurs the

far-infrared becomes semi-transparent (e.g., Cox et al. 2015). Under such conditions, instrument-dependent biases of -2 to -6 Wm$^{-2}$ in pyrgeometers have been reported (Gröbner et al. 2014).

Uncertainty in SWD is plotted Figure 3b as a function of SZA. In addition to absolute units, the uncertainty is also shown in relative units (%). The BSRN target uncertainty for pyranometers is 2% (McArthur 2005), a condition that is met, and the uncertainty is relatively flat when the SZA is < 70º. When the SZA is larger, the relative uncertainty is larger too, but

the absolute uncertainty is < 5 Wm$^{-2}$, which meets the standard for the minimum expected error (McArthur 2005).

## 3. Biases caused by ice

To better understand the consequences of icing, Figure 4 shows a case study for LWD from late January in panel a

and for SWD on 13-14 April in panel b. Analysis of these cases, next, is followed in Section 3.3 with a more general calculation of biases at the monthly scale.

### 3.1. LWD icing case

The LWD time series spans approximately two weeks and shows a range of LWD typical of the Arctic winter, from ~140 Wm$^{-2}$ during the coldest, clearest times to ~280 Wm$^{-2}$ during the warmest, cloudiest times. The blue line in Figure 4a is the LWD BE and the grey shading is the uncertainty. The red line is the time series of pyrgeometer 28507, which was susceptible to icing. The red shading highlights the bias caused by the ice in 28507 relative to the BE. Two events stand out: First, beginning on 22 January, was a frost event that completely covered 28507, shown in the inset image. The effect on 28507

can be compared to the clear dome in the neighbouring pyrgeometer (34309), which was included in the BE. Later, on 27 January, a rime event caused by freezing fog occurred with similar results for the pyrgeometers. For both events, the biases were +40 to +45 Wm$^{-2}$, with the maximum observed bias during the case study of +58 Wm$^{-2}$ occurring on 25 January. (This is comparable to the maximum bias from icing observed during the campaign of +59.5 Wm$^{-2}$ on 26 December, though occasionally larger biases were observed briefly in association with melting ice or snow on the dome.) The systems in the

figure are identical except that the radiation shield covering 34309 was lifted 2 mm using washers to improve airflow over the dome, highlighting the influence of small differences in ventilator design. These cases were not the only periods of severe



icing during D-ICE, but the example shown was unusually impactful on 28507 and is therefore highlighted because it is instructive. The bias is largest when the sky is clear. This is because when the sky is clear, the sky emissivity (and thus brightness temperature) is most different from that of the ice. When clouds are present, the bias is reduced to near 0 Wm$^{-2}$.

This is evidenced by the brief occurrence of an optically-thick stratiform mixed-phase cloud with a liquid layer between 2 and 3 km late on 23 January, highlighted in the top-centre panel of Figure 4a. While this cloud was over the station, the icing conditions remained unchanged, as confirmed by review of the camera images (Figure 4a centre inset), but the flux from the ice was similar to that of the cloud because both had similar emissivity and thermodynamic temperature (as confirmed by radiosoundings, not shown). Thus, this case study demonstrates that biases in LWD can be large, but are episodic in nature,

depending on both ventilator performance and cloud properties, in addition to icing conditions. The bias in 28507 relative to the BE averaged for the time period represented in the figure is +7.8 Wm$^{-2}$, which is consistent with the biases reported during brief periods having icing conditions elsewhere (Persson et al. 2018).

### 3.2. SWD icing case


Figure 4b shows an example of a clear-sky day in mid-April that followed frost formation the previous night. The deposition period of the frost ended at approximately 17:00 UTC on 14 April after which the frost sublimated during the day. The black line in the figure shows the SWD BE and the grey shading is the uncertainty. Five pyranometers that had frost on their domes for at least some of the day are shown by the coloured lines. Missing data in the figure are because of shadows.

The biases from the ice are generally positive (up to +188 Wm$^{-2}$), but sometimes also negative (up to -106 Wm$^{-2}$) and the maximum absolute and relative errors were not necessarily coincidental in time. These values are comparable to the largest instantaneous biases observed during the campaign, though several occurrences on other days (e.g., 28 April) of negative biases as large as -211 Wm$^{-2}$ were associated with thick pieces of ice and snow that accumulated on the lee side of the dome and then later faced the direct sun. In Figure 4b, some biases (e.g., 160002) manifest as an apparent shift in the solar cycle, but

the data appears otherwise physical. Note that the thickness and density of the ice in the example images (insets in figure) is qualitatively similar. The main difference between the images is in the coverage of the ice. The dome of both the pyranometers (32421, F16305R) that exhibited negative biases during the day were entirely covered by ice when the negative biases were observed; the bias was negative because the irradiance was attenuated. In contrast, the ice on the domes of the other affected radiometers (and also F16305R and 32421 later in the day) was concentrated near the top of the domes. This is a condition

commonly observed on ventilated radiometers that is colloquially known as "capping". Capping is thought to be the consequence of the ventilator being least efficient in circulating air over the top of the dome; thus, deposition typically occurs at the top of the dome first and ice in this location also takes longer to sublimate. The contrast between the negative bias during the early and middle parts of the day and the biases of the capped radiometers, including F16305R and 32421 as they began to sublimate later the day, is notable: While not definitive, it suggests that capping tends to result in positive biases, at least during

relatively low sun angles (the maximum solar elevation angle for this case was 28.3º) when the direct beam passes largely



uninhibited through the clear portion of the dome and the signal is enhanced by scattering towards the thermopile by the ice at the top of the dome.

To illustrate the influence of ice during diffuse conditions, another example from 7 April (Figure 5) shows the transition from a positive bias (dominated by scattering) to a negative bias (dominated by attenuation) in F16305R capped with ice due to a transition in lighting. From 20:00 to 20:30 UTC the direct beam is present, being 10-30% of the total irradiance, during partly cloudy conditions. At that time, a bias of up to +50 Wm$^{-2}$ is observable and after which the bias becomes negative, about -25 Wm$^{-2}$, as overcast conditions obscure the sun and the lighting becomes entirely diffuse.

### 3.3. Biases from ice in monthly means

Figure 6 shows monthly mean biases in LWD (panel a) and SWD (panel b) for each radiometer. For each cell, the colour indicates the bias associated with frost, rime, snow, or liquid (usually ice melted by heat from the ventilator). The monthly means are also plotted as a time series in panels c and d with the aggregate means shown by solid lines. Note that the months of July and August include limited amounts of data because of beginning and end dates of the campaign. All bias calculations are corrected to account for differences between individual radiometers and the BE that is associated with calibration uncertainty. The bias calculations are insensitive to the determination of ice occurrences from the images because the average of all conditions, regardless of ice presence, is calculated.

As noted in Section 3.1, the LWD case study was chosen because it was a particularly influential event and a particularly susceptible system was highlighted. Figure 6 shows that when data are averaged for long periods of time, the bias in icing of pyrgeometers is actually small. Indeed, only two radiometers, BSRN and 28507 (having similar configurations and equipment), exhibit biases that are detectable relative to the average uncertainty (Figure 3). The most severely affected month was January when the average bias was just +1 Wm$^{-2}$ and the most affected system was biased +4 Wm$^{-2}$. Note that some experiments were conducted during icing periods on four days (82 hours total) in the first week of January (described later in Section 4) and that these times were rejected from the analysis in the figure.

SWD icing biases during D-ICE occurred from February through June with a peak in April. This is because biases in SWD depend both on the amount of sunlight and the amount of icing, which have opposing seasonal cycles. The opposition is slightly out of phase such that in autumn there was too little sunlight when the icing first began in earnest, but that both substantial amounts of sunlight and icing co-occurred during spring. Recall from Section 2.2 that the beginning of the icing season was late during D-ICE and that in a more typical year at Utqiaġvik some biases may also have been observed in September and October. Note that the calculation includes the average of both negative and positive biases. If the average of the absolute value of the bias is plotted instead (not shown), the biases increase slightly but interpretation is hampered by the fact that noise contributes to the bias calculation rather than cancelling out. Nevertheless, the results indicate that biases in pyranometers at D-ICE were dominated by positive perturbations, which is consistent with spurious data being principally tied to a combination of clear-skies, low sun angles and capping in early spring.




## 4. Ice Mitigation

### 4.1. Performance of ventilators

To assess ventilator performance, we begin with two qualitative examples that illustrate broadly the influence that heating and ventilation have in mitigating ice. The first example is of a freezing fog event that occurred from 1230 UTC on 5 January with rime accumulation continuing until about 0900 UTC on 6 January. The image in Figure 7a show the status of the systems during the event at 1900 UTC, 5 January. At this time, rime is observable on the domes of some of the systems while most remain ice-free. Immediately after these images were taken, the power was deliberately cut to the ventilators and the

radiometers began accumulating ice immediately, being iced over within 2.5 hours (Figure 7b). The second example began on 2130 on 9 January when the RHI > 100%, but no active icing was observed, the instruments were intentionally iced by manually spraying water on the domes and de-icing was monitored for 9 hours before the experiment was ended by the onset of a natural riming event. Of the 25 tested systems, 17 successfully de-iced within the 9-hour window in the supersaturated conditions (mean 6 hours, minimum 0.25 hours). The systems that de-iced the fastest were those that featured heating elements, though

several unheated systems were observed to de-ice themselves.

To quantify performance over the course of the campaign, Figure 8 shows a summary of statistics from the systems at D-ICE, NSA and OLI based on the classification of the images described in Section 2.4.2. The systems are labelled on the x-axis and the y-axis shows a simple performance metric, which is calculated thusly:

$$P = 100 * \left\{ \left[ \frac{t_{i,iced}}{t_{icing}} \right] - 1 \right\}, \tag{1}$$

where $P$ is the ice mitigation performance in units of percent, $t_{i,iced}$ is the amount of time system $i$ was iced, and $t_{icing}$ is the amount of time icing conditions occurred. Therefore, when $P = 0\%$, the amount of time the radiometer was iced and the amount of time icing occurred are the same; i.e., the de-icing system had no effect. When $P < 0\%$, the value expresses the percent of time during icing conditions that the system successfully mitigated ice, with a minimum of -100% (all icing mitigated) because $t_{i,iced}$ cannot be less than 0. Instances where $P > 0\%$ indicates that the radiometer was iced more frequently than icing conditions

occurred, suggesting that the ventilator exacerbated icing. Positive values of $P$ can theoretically reach infinity because $t_{icing}$ is independent of, and can pose no restriction on $t_{i,iced}$ (for example, over some time $t$ if no icing conditions are observed but radiometer $i$ was iced then $t_{icing} = 0$, $t_{i,iced} > 0$ and $P = \infty$).

Uncertainty in $P$ arises from errors in $t_{i,iced}$ or $t_{icing}$. Recall from Section 2.4.2 that the identification criteria for the status of the individual radiometers ($t_{i,iced}$) is that all ice is flagged, regardless of coverage, density, or thickness. This criterion

is advantageous because it is an objective threshold for icing but also has the disadvantage that instances of very light icing could be missed during classification. To attempt to understand the uncertainty in $t_{i,iced}$, we compared systems that were treated most similarly (the 2 MS80s, the 2 SW CVF4s, the 2 LW CVF4s and 2 VEN/PSP systems). We found that the differences



were between 2% and 20%, averaging to 8.5%. We acknowledge that we do not have a robust way to calculate this uncertainty, but it is likely conservative because while some of the observed difference reflects error in the classification, the treatment of

the compared systems was not precisely the same nor can local variability in icing be ruled out, and upon review some differences between similar systems are found to be real (see below). Identification of events ($t_{icing}$) is subjective and uncertain as well, in particular in determining the end time for an event. However, uncertainty in $P$ due to uncertainty in the $t_{icing}$ is a constant applied to all radiometers. Consequently, the magnitudes of $P$ in Figure 8 may be biased, but if so are biased uniformly while the relative differences in $P$ between the systems have uncertainties of approximately 10% based on the comparison

between like systems. Particular caution should be exercised in interpreting the differences between the sites, D-ICE, NSA and OLI because $t_{icing}$ was determined independently for each.

While no systems were found to be 100% effective, two-thirds of the systems, including all those housing pyrgeometers, were effective at least 80% of the time and 15/34 were effective at least 90% of the time. The average was 77%, but there was also a substantial amount of variability (σ = 30%) and five systems were found to mitigate ice < 50% of the time.

One of these was observed to increase icing. The positive value of $P$ for this system, F16305R, is consistent with subjective analyses that indicated ice (likely frost) on the F16305R's dome when no ice was present on the ventilator or nearby structures. While we do not know the explanation for this behaviour, the most noticeable design element of F16305R's ventilator (which was unheated) is that the aspiration vent around the base of the dome is larger than any of the other systems. It is plausible that this could produce a low-pressure pocket around the dome that could support deposition.

Interestingly, the CMP22 outperformed the SMP22 in the CVF4 by 20% despite the similarity between the ventilation systems and the radiometers. The only difference was that on 6 January the air intake screen on the CVF4 holding the SMP22 was removed to assess whether clogging by snow and reduced air flow impacted effectiveness. The CMP22 was observed to outperform the SMP22 by 19% prior to this change and 21% after, so the difference is not attributable to the presence of the screen and the screen apparently had little impact on effectiveness. We do not know the explanation for the observed difference.

In general, mitigation of ice on pyrgeometers was more effective than pyranometers, even for cases when the systems were otherwise similar (e.g., CVF4s and VENs). There are several plausible, but not mutually-exclusive explanations for this: First, the domes of the pyrgeometers are smaller and lower-profile, and therefore aspirated air may be more easily circulated to the top of the dome; second, the smaller surface area of the dome supports improved conduction of heat, as does the fact that pyrgeometer domes are constructed of silicon, which is more thermally conductive than the quartz pyranometer domes;

and third, more speculatively, the outer coatings of the domes may be less prone to accretion of ice.

## 4.2. Physical Mechanism

Successful mitigation of ice is demonstrated by systems in Figure 8 that were not equipped with heaters. This supports

the heuristic within BSRN that ventilation of ambient air alone can be effective. However, it is counter-intuitive because



aspiration of saturated air increases rather than decreases deposition rates, specifically resulting in denser, but not necessarily thicker, frost (Kandula 2011 and references therein).

We examined the properties of the Eppley ventilation system configured similarly to those in use at the Barrow BSRN station to help elucidate the attributes that contribute to effectiveness in the absence of heating elements. The tested system is
an Eppley VEN housing a high-flow 80 cfm (10.3 W) DC fan (Delta Electronics FFB0812EHE) modified with bearings rated for low-temperature; examples of such systems at D-ICE are in positions 6-9 and 24 (Figure 1b). When outside of the ventilator, the velocity of the air downstream of the fan is ~9.8 m s⁻¹. When installed in a standard VEN configuration, the maximum velocity measured near the top of the dome is 7.7 m s⁻¹. When the shield is lifted to improve airflow, the velocity increases to 8.6 m s⁻¹ for a 1 mm lift and to 9.3 m s⁻¹ for a 2 mm lift (as in positions 6 and 8).

Figure 9a shows a 9-hour time series of temperatures collected during D-ICE in January, 2018. The dome temperatures from PIRs 28507 and 34309 were 0.5-0.6 °C warmer than the ambient air temperature and differed from each other by about 0.15 °C. On 5 January both fans were shut down simultaneously and the dome temperatures agreed after equilibration to ambient conditions (difference 0.01 °C RMSE). Thus, the 0.15 °C difference represents a real difference in temperature and not uncertainty. We will return to this later.

The fan in 34309 was turned off shortly before 20:00 UTC on 8 January for ~30 min before being turned on again. It was then turned off a second time at ~21:00 UTC for ~60 min. As observable in Figure 9a, in both cases the dome temperature decreased ~0.6 °C to the ambient 2 m air temperature measured by the station. The fan in 28507 ran continuously during the experiment and its dome temperature responded only to changes in meteorology. At -25 °C, a 0.6 °C increase in temperature subsaturates air (w.r.t. ice) with an RHI as high as 105%, which is a higher supersaturation than typically occurs. Thus, the
heat added by the fan is sufficient to explain the effectiveness of the ventilators without heating elements.

To better understand the sources of the heating, the experiment was repeated under controlled conditions in a laboratory in Boulder, Colorado. First, an FFB0812EHE was placed in a cold chamber without the VEN and a thermocouple (Type-T; copper-constantan) was positioned in the air stream ~10 cm from the fan. The apparatus was allowed to equilibrate to -15 °C for several hours without the fan running after which the fan was started using an external control and a temperature
increase of 0.35-0.4 °C was observed over the course of several minutes after which the fan was turned off again and the temperature returned to its previous value (purple in the Figure 9b). A similar result was observed using a stock ebm-papst 8212JN that has comparable specifications to the FFB0812EHE (yellow in Figure 9b). This test confirms that the temperature increase observed at D-ICE was supported by the fan independent of the ventilator.

Next, the experiment was repeated again using an FFB0812EHE installed in a VEN containing a PIR and having a
shield with a 1 mm lift and clay sealing around the shield edges. Similar to the previous iteration, a 0.39 °C increase in dome temperature occurred (blue in Figure 9b). The case temperature increased by the same amount (red in Figure 9b), slightly lagging the dome because the case thermistor is more deeply embedded within the radiometer's mass. Both case and dome equilibrated within 1 hour. After the fan was turned off, the system re-equilibrated to within 0.05 °C of its unperturbed state



after 3 hours. Finally, the experiment was repeated two more times with changes to the height of the lift of the radiation shield

(thin blue lines in Figure 9b). When the shield was lifted to 2 mm, the equilibrated temperature increased relative to the 1 mm lift and when no lift was provided the smallest temperature increase was recorded. This is consistent with the results shown earlier, which indicate that system with the lifted shield had a warmer dome (Figure 9a) and performed better overall during D-ICE (Figures 4, 6, and 8).

The heating of the dome by the fan is principally from two sources. The first is heating of the air moving past the fan

motor, which is warmed by waste energy. This can be calculated by first estimating the amount of waste heat in Watts, $H$, from the static pressure reported by the manufacturer, $S$, and the observed volume, $V$, moved by the fan in $t$ seconds,

$$H = 1 - \frac{SV}{t},$$ (2)

and then finding the temperature increase, $T_w$, associated with $Ht$ Joules applied to the same volume,

$$T_w = \frac{H}{c_p \rho V},$$ (3)

where $c_p$ is the specific heat of the air at constant pressure, and $\rho$ is the air density. Note that Eq. (3) neglects energy that is radiated or conducted away.

The second source is that the air downstream immediately in contact with the radiometer necessarily comes to rest and thus undergoes an adiabatic compression. This topic has been studied extensively for high velocity flows (e.g., Thompson 1968; Lenschow 1972), but less at low velocities. At low velocities, the properties of the gas can be approximated as ideal.

Therefore, we formulate the problem from the first law of thermodynamics beginning with the ideal gas law, differentiating as follows:

$$dT = \frac{\alpha \, dP + P \, d\alpha}{R},$$ (4)

where $T$ is the temperature, $\alpha$ is the specific volume of the air (i.e., the inverse of the air density), $P$ is atmospheric pressure, and $R$ is the gas constant. The calculation is insensitive to the reduced density of the humid air and it is sufficient to set $R =$

$R_d$. The system is assumed to be in hydrostatic equilibrium such that $dP = 0$, and so Eq. (4) reduces to

$$dT = \frac{P \, d\alpha}{R}.$$ (3)

$d\alpha$ is found as follows:

$$d\alpha = \alpha_0 - \alpha_{tot},$$ (4)

where $\alpha_0$ is the specific volume of the air at static atmospheric pressure, $P$, and $\alpha_{tot}$ is the specific volume of the total pressure,

the sum of the static and dynamic pressures; i.e., it is the change in specific volume caused by the compression of the air where it comes to rest at the surface of the radiometer downstream of the fan. Note that here, $dT$ is treated as idealized, but that in actuality the adiabatic compression is not perfectly efficient.

The results (Figure 9c) indicate that for the experimental setup in question, waste heat and adiabatic heating contribute similarly to the observed temperature increase in the chamber, with ~44% (0.17 °C) explainable by the former and ~42% (0.16





°C) explainable by the latter at velocities corresponding to the ventilator. The remaining 14% was unaccounted for, but besides error may be associated with secondary effects such as drag (Lenschow, 1972) and turbulence.

The lab experiments resulted in about half of the total heating that was observed in the January D-ICE case. D-ICE was carried out at sea level whereas the tests in Boulder (~1500 m.a.s.l.) were carried out at lower (~15%) atmospheric pressure. The field case study shown in Figure 9a was also colder (-25 °C) than the chamber tests (-15 °C). Both of these factors increase
the air density and therefore reduce the calculated temperature change. Thus, differences in the state variables do not explain the difference in fan heating observed between the field and the lab. The fan speed was monitored at D-ICE and did not indicate that fan efficiency was lower than the lab because of clogging by snow. Instead, we hypothesize that the difference could plausibly be affected by wind forcing, which also results in a compression of the air stream and subsequent heating (e.g., Thompson 1968), in this case at the point of impact with the dome.

Interestingly, that heating from waste heat increases as air velocity is decreased (Figure 9c) counters the observed relationship between increasing the lift of the shield and increases in both air velocity measured at the top of the dome and dome temperature. This suggests that the effect of raising the shield is to enhance the circulation of air around the dome.

## 5. Discussion and Conclusions


The De-Icing Comparison Experiment (D-ICE) was carried out in 2017 and 2018 at the NOAA Atmospheric Baseline Observatory in Utqiaġvik, Alaska (71.3˚N) in collaboration with the DoE ARM NSA (also Utqiaġvik) and Oliktok Point (250 km east of Utqiaġvik) stations. D-ICE collected data suitable for assessing technology designed to mitigate the formation of ice on broadband radiometric stations and to quantify the influence of ice on the flux measurements. Over the course of an
Arctic cold season, the status of icing on a total of 34 upward-facing broadband radiometers was monitored using cameras. Most of the radiometers were housed in ventilators that aspirated air over the sensors, sometimes heated, while others were designed with internal heating and/or ventilation. The systems were contributed by research institutes and commercial vendors and were representative of the types used by research-grade programs, such as the BSRN and DoE-ARM.

System performance, defined as the amount of time a radiometer was classified as iced normalized by the amount of
time icing conditions were present (Eq. 1), was in the mean amongst the systems, a 77% reduction in the expected amount of ice. Thus, on average the systems tested during D-ICE were successful in mitigating most ice. Ice was more effectively mitigated from pyrgeometers than pyranometers. Many systems housing either type of radiometer were 90% effective or better, including some that did not use external heat. Even systems without external heating elements were observed to have radiometer domes that were warmer than ambient air by 0.5 to 0.6 °C, which is sufficient to subsaturate the air (w.r.t. ice)
during typical icing conditions, explaining the skilful performance. Through field and post-campaign laboratory tests, the source of this heating was found to have approximately equal contributions from waste heat from the (~10 W) ventilator fan and adiabatic heating from compression of the air downstream from the fan. Thus, while heating elements were found to be





effective, they are not required for successful ice mitigation. Instead, an important factor for success appears to be effective circulation of air over the dome.

Generally, we did not identify significant errors caused by the ventilators and the night time offsets in all systems were small, consistent with Michalsky et al. (2017). One exception was heated ventilators that were susceptible to clogging by snow. These were observed to have small night time offsets correlated with wind velocity but not net longwave, the latter being expected for errors from infrared loss (Dutton et al. 2001). Instead, we postulate that blowing snow clogged the ventilators, reducing aspiration and causing differential heating of the radiometer after which during calm winds the heated

ventilator unclogged the inlet.

       When ice was present on sensors, the instantaneous biases varied, but could be large, up to +60 Wm$^{-2}$ in the LWD and from -211 to +188 Wm$^{-2}$ in SWD. However, the monthly mean biases from ice were found to be similar to, or smaller than uncertainties from other sources, which is approximately 3 Wm$^{-2}$ (LWD) and 2% (SWD). There are a few reasons for this unexpected result. First, the fact that icing conditions were not present continuously was a factor: from November through

April, icing conditions occurred 63% of the time. The skilful performance of the ventilators was also a notable contributing factor in reducing the bias, but there were other factors as well. For SWD, compensation between positive and negative biases provided some cancellation in the monthly mean. The positive biases were tied to scattering of the direct beam when pyranometer domes had a cap of ice at the dome top at low sun angles during clear skies. Negative biases by attenuation when domes were completely encased in ice were less common. While negative biases appear to dominate during diffuse (cloudy)

conditions, the irradiance is also less at those times, which reduces the bias. Overall, positive biases were found to dominate for SWD and the cancellation did not explain the relatively small monthly mean errors. Instead, it is the timing of the occurrence of icing that likely explains the result: biases in SWD in the monthly mean were only found in spring when both sufficient sunlight and sufficiently frequent icing conditions co-occurred. Icing in spring is commonly in the form of diurnal (morning) frost, which occurs during relatively low-light times of the day; i.e., the icing is most severe when the irradiance is

small. Therefore, we recommend scheduling daily maintenance in early morning during spring for efficiency. Indeed, icing was most severe during the polar night (November – January) when the net radiation is dominated by the longwave and pyranometers do not measure any signal. It is thus fortuitous that pyrgeometers were more effectively kept ice-free than pyranometers; i.e., small monthly mean icing biases can partially be attributed to the fact the when icing is most severe, radiometric measurements rely on a type of sensor that is more easily kept free of ice. LWD biases from ice were large when

the sky was clear, but reduced to near-zero in overcast conditions. Thus, the fact that the frequency of cloud occurrence is near 80% in the annual mean at Utqiaġvik (Shupe et al. 2011) contributed to reducing LWD biases. This was also a mitigating factor for the SWD because the largest biases were observed near solar noon on clear days. Since cloud amount, icing meteorology, and available sunlight are all important, geographical location is meaningful and the results presented here may not be representative of other Arctic locations. We hypothesize that this may be true also for locations in the vicinity of

Utqiaġvik, such as over the sea ice where local or mesoscale natural de-icing mechanisms (such as downslope wind events) may be more limited.



Consistent with earlier studies reporting difficulty in distinguishing iced data in post-processing (Lanconelli et al. 2011; Matsui et al. 2012), we find that quality control procedures are poorly-suited for detection of iced data because the signal caused by ice is not statistically outside the range of variability in the signal caused by clouds. Therefore, common screening methods (e.g., Long and Shi 2008) are insufficient. Some of the non-definitive tests proposed by Long and Shi that involve cross-comparison between sensors may be more likely to identify suspect data, but results are dependent on the differential icing characteristics between the sensors. Other tests have been proposed such as comparing the sign and time derivative of the difference between upward and downward LW and SW measurements (van den Broke et al. 2004; Wang et al. 2018), though these tests rely on similar assumptions. Some studies rely on logbooks from station personnel and thresholds for relative humidity (e.g., Sedlar et al. 2011; Miller et al. 2015, 2017; Persson et al. 2018), but because of the relative infrequency of observer records (e.g., daily) and suspect reliability of RHI as a proxy for icing (Section 2.2), these methods also have limitations. D-ICE demonstrates success in quality control by monitoring instrumentation with cameras, but this approach is not always practical. In keeping with van den Broke et al. (2004), we suggest that time-derivative analysis for ice detection should be further explored. For example, the variability in the iced data in the SWD case (Figure 4b) is much slower and smoother than would be expected from clouds in a regime not dominated by the diffuse. Thus, development of new algorithms that flag iced data based on time-variant tests might be possible if the regime can be determined to be dominated by the direct beam and can be distinguished from the consequences of instruments being out of level or expected differences between the global and SUM SWD though the day, both of which can produce structurally-similar errors.

Finally, as a baseline for comparison used for analysis, a "best-estimate" data set was produced using a combination of the measurements that were ice-free. Though an unexpected outcome of D-ICE, the number of radiometers, variety of systems and skilful performance of the systems resulted in production of a verified ice-free data set that is nearly 100% complete for the duration of the campaign. Empirically-based uncertainties were also calculated from the variability amongst the ice-free observations. This data set is uniquely well-characterized in the Arctic and therefore may be suitable for use beyond inquiry related to ice-mitigation. For example, D-ICE took place during the Year of Polar Prediction (YOPP) at one of the YOPP "supersite" observatories and the campaign spanned the wintertime YOPP Special Observing Period (SOP) #1 during February and March 2018. We therefore propose that the D-ICE best-estimate data products (Cox, 2020b) may be useful for model evaluations, such as the planned YOPP site Model Intercomparison Project (YOPPsiteMIP).

**Acknowledgements**

This work is dedicated to the memory of the late Charles N. Long (formerly of CIRES) who first suggested the concept for D-ICE and provided invaluable guidance and support in developing and carrying out the experiment. We appreciate deployment assistance from J. Booth (NOAA), N. Lewis (Univ. Colorado), M. Helmberger (Univ. Colorado), C. Schultz (NOAA), A. Clarke (NOAA), A. Looze (NOAA Pathways intern), K. Olivas (NOAA summer intern); field support from D. Oaks (Fairweather LLC), B. Bishop (Sandia), W. Brower (UIC Science, retired); engineering and equipment support from R. Albee



(formerly STC); logistical support from B. Vasel (NOAA), J. Mather (PNNL), M. Ivey (Sandia), F. Helsel (Sandia), M. Steufer (Univ. Alaska, Fairbanks), and useful conversations with R. Zamora (NOAA, retired), who assisted with adiabatic heating calculations; P.O.G. Persson (CIRES), who suggested that natural de-icing processes vary geographically; G. König-Lango (AWI, retired), J. Osborn (CIRES), M. Shupe (CIRES); M. Martinsen (NOAA) who provided science and logistical guidance; R. Lataitis (NOAA) who provided an internal review; and guidance from members of BSRN's Cold Climates Issues Working Group (CCIWG). The following organizations (points of contact) contributed equipment to the campaign: Delta-T (Dick Jenkins), Kipp & Zonen (V. Cassella), Hukseflux (J. Konings), Eppley (T. Kirk), EKO (W. Beuttell), PMOD/WRC (J. Gröbner), Environment and Climate Change Canada (ECCC, Andrew Platt), NOAA, MeteoSwiss (L. Vuilleumier), NCAR (S. Oncley, S. Semmer, K. Knudeson) and the Alfred Wegener Institute (H. Schmithüsen, B. Loose). Images collected by ARM as part of DICEXACO are available from the ARM data archive (https://www.arm.gov), https://doi.org/10.5439/1507148. NOAA-GML station data are available from https://www.esrl.noaa.gov/gmd/obop/brw/. D-ICE data are available from NOAA-NCEI: Cox (2020a, https://accession.nodc.gov/0209059) processed radiometric data and Cox (2020b, https://accession.nodc.gov/0209058) best-estimate files. All D-ICE images and raw data files are available from NOAA-PSL through the D-ICE web portal, https://www.esrl.noaa.gov/psd/arctic/d-ice/. QCRAD radiometric data products for OLI and NSA are available from the ARM archive, https://doi.org/10.5439/1027372, ceilometer data from https://doi.org/10.5439/1181954, and radiosoundings from https://doi.org/10.5439/1021460. D-ICE was supported by the NOAA Arctic Research Program, the NOAA Physical Sciences Laboratory of the Earth Systems Research Laboratories and the U.S. Department of Energy (DoE) Atmospheric Systems Research (ASR) grant DE-SC0013306. Campaign logistical support was provided by the NOAA Global Monitoring Laboratory. This work was authored in part by the National Renewable Energy Laboratory, operated by Alliance for Sustainable Energy, LLC, for the U.S. Department of Energy (DOE) under Contract No. DE-AC36-08GO28308. Funding provided by U.S. Department of Energy Office of Energy Efficiency and Renewable Energy Solar Energy Technologies Office. The views expressed in the article do not necessarily represent the views of the DoE or the U.S. Government. The U.S. Government retains and the publisher, by accepting the article for publication, acknowledges that the U.S. Government retains a nonexclusive, paid-up, irrevocable, worldwide license to publish or reproduce the published form of this work, or allow others to do so, for U.S. Government purposes.

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

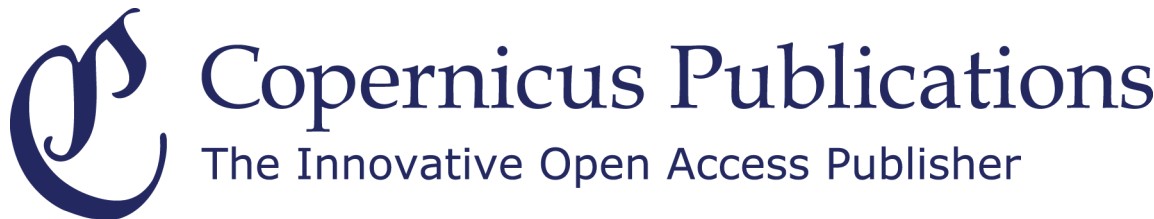








**Table 1.** *List of quality control procedures received by D-ICE instruments. "x"s denote the procedure was implemented and "o"s indicate the procedure was not implemented.*

| Table Position | Radiometer Serial # | Band | Calibration Source | Table Illumination Removed | QC: Visual Screening | QC: Long and Shi (2008) | QC: Shadow Flag | QC: Icing Flag | QC: IR-Loss Correction | QC Notes |
|---|---|---|---|---|---|---|---|---|---|---|
| 1 | 34231F3 | SW | GMD | x | x | x | x | x | x | None |
| 2 | 160478 | SW | GMD | x | x | x | x | x | x | None |
| 3 | 160183 | LW | GMD | o | x | x | o | x | o | None |
| 4 | 160002 | SW | Factory | x | x | x | x | x | o | None |
| 5 | 160008 | LW | Factory | o | x | x | o | x | o | None |
| 6 | 26818F3 | SW | GMD | x | x | x | x | x | x | None |
| 7 | 18135F3 | SW | GMD | x | x | x | x | x | x | None |
| 8 | 34309F3 | LW | GMD | o | x | x | o | x | o | None |
| 9 | 28507F3 | LW | GMD | o | x | x | o | x | o | None |
| 10 | F16305R | SW | GMD | x | x | x | x | x | x | None |
| 11 | 26214 | SW | GMD | x | x | x | x | x | x | No cleaning; Poor Level |
| 12 | 130814 | SW | GMD | x | x | x | x | x | x | Fan dead 27-Dec-2017 to 04-Jan-2018 |
| 13 | A1571 | SW | Factory | x | x | x | x | x | o | Ventilator shadowing |
| 14 | 20523F3 | SW | GMD | x | x | x | x | x | x | Fan dead 19-Jan-2018 to 08-Feb-2018 |
| 15 | 38172F3 | SW | GMD | x | x | x | x | x | x | None |
| 16 | 26236 | SW | GMD | x | x | x | x | x | x | None |
| 17 | 130819 | SW | GMD | x | x | x | x | x | o | None |
| 18 | 4037 | LW | Factory | o | x | x | o | x | o | None |
| 19 | S16088025 | SW | GMD | x | x | x | x | x | o | None |
| 20 | S16090016 | SW | GMD | x | x | x | x | x | o | None |
| 21 | 2510 | SW | GMD | x | x | x | x | x | o | None |
| 22 | A1338 | SW | Factory | x | x | x | x | x | o | None |
| 23 | 2060 | SW | Factory | x | x | x | x | x | o | None |
| 24 | 8041 | SW | GMD | x | x | x | x | x | o | None |
| 25 | 130617 | SW | GMD | x | x | x | x | x | o | None |
| 26 | 970426 | SW | GMD | x | x | x | x | x | x | None |








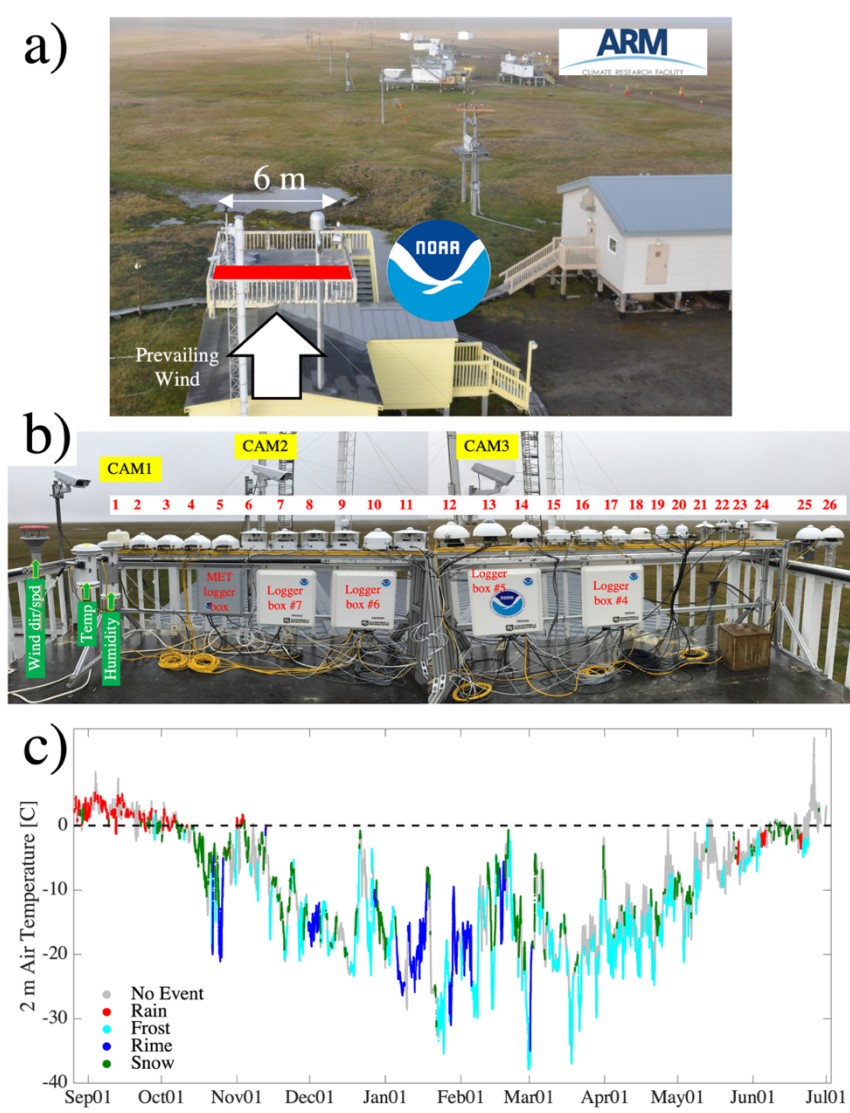

**Figure 1.** *(a) NOAA-GML Barrow Atmospheric Baseline Observatory in Utqiaġvik, Alaska (71.325º N, 156.625º W, 8 masl), DoE-ARM facility in the background (image from https://www.esrl.noaa.gov/gmd/obop/brw/). The red square in (a) shows location and orientation of the D-ICE table, which is pictured in (b). (c) Time series of air temperature color-coded with the types of precipitation and icing events that were recorded: rain (red), frost (cyan), rime (blue), snow (green), and gray (no event).*





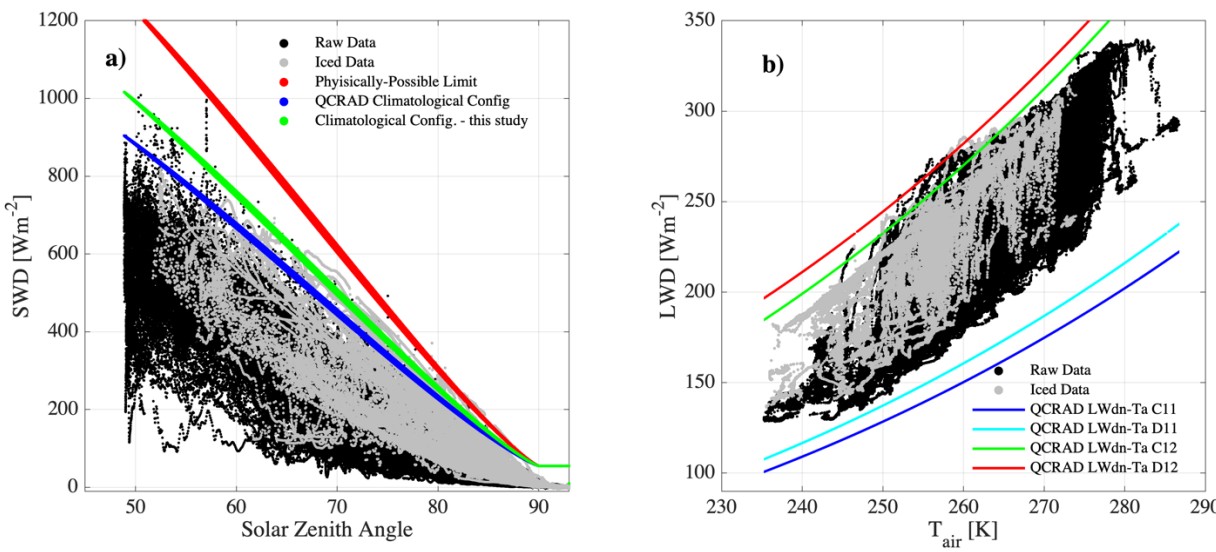

***Figure 2.*** *Examples of climatological configurable limit tests from Long and Shi (2008) for SWD (SN F16305R) (a) and LWD (SN 28507) (b). The figures are comparable to figures 1 and 9 from Long and Shi except that 1 min average are shown here instead of 15 min. Black points are ice-free and gray points are contaminated with ice. The colored lines are suggested thresholds for rejection of data of varying degrees if strictness.*







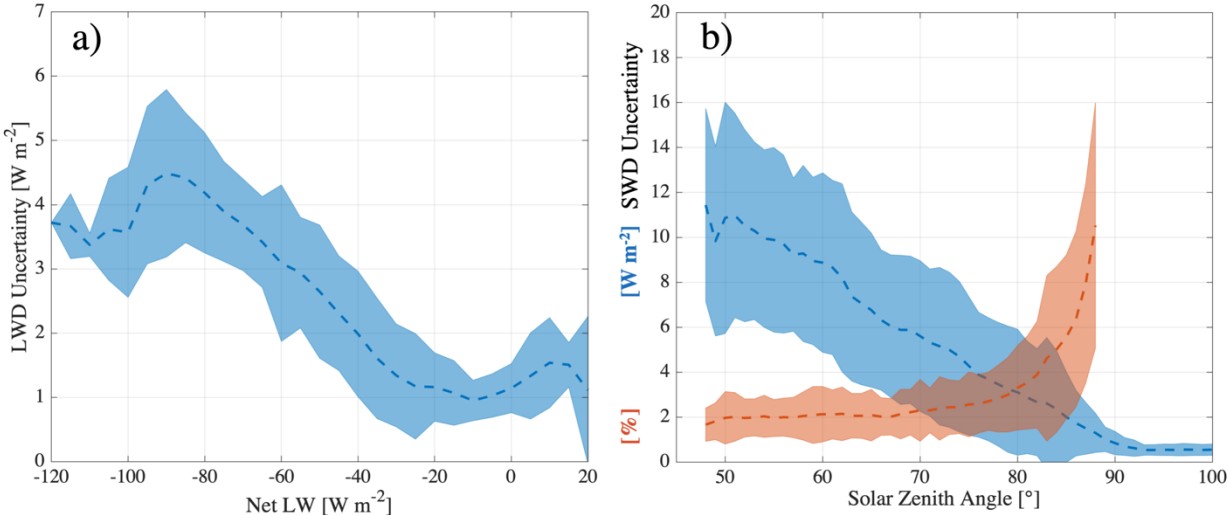

**Figure 3.** *(a) Mean (dashed line) and 1σ (shading) variability in calculated 1 min average uncertainties for LWD as a function of the net longwave flux represented by the mean thermopile flux of the pyrgeometers. (b) Similar to (a) for uncertainty in SWD plotted against solar zenith angle. Uncertainties in (b) are plotted in units of W m² (blue) and relative units (red).*



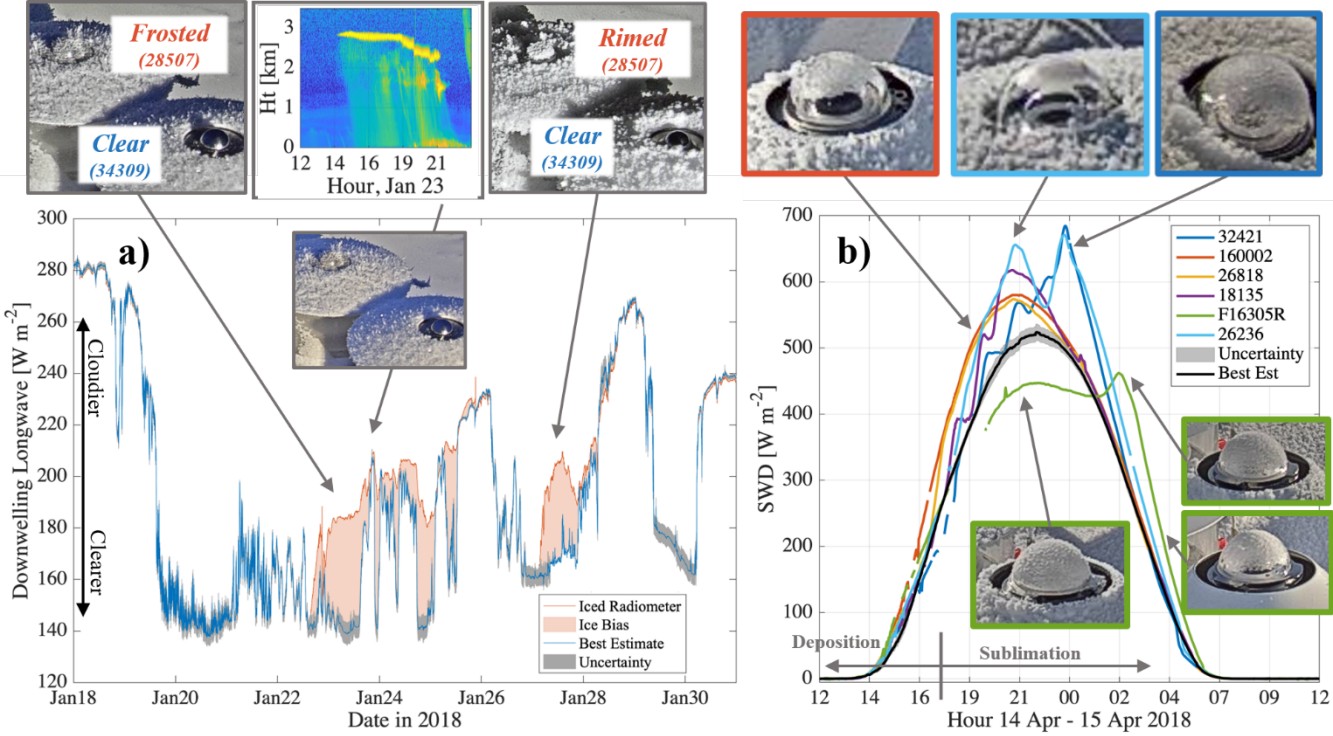

**Figure 4.** *Case studies of icing for LWD in late January (a) and SWD during a clear day in April (b). Inset images show condition of highlighted radiometers. The upper-center panel in (a) shows backscatter from the DoE-ARM ceilometer.*

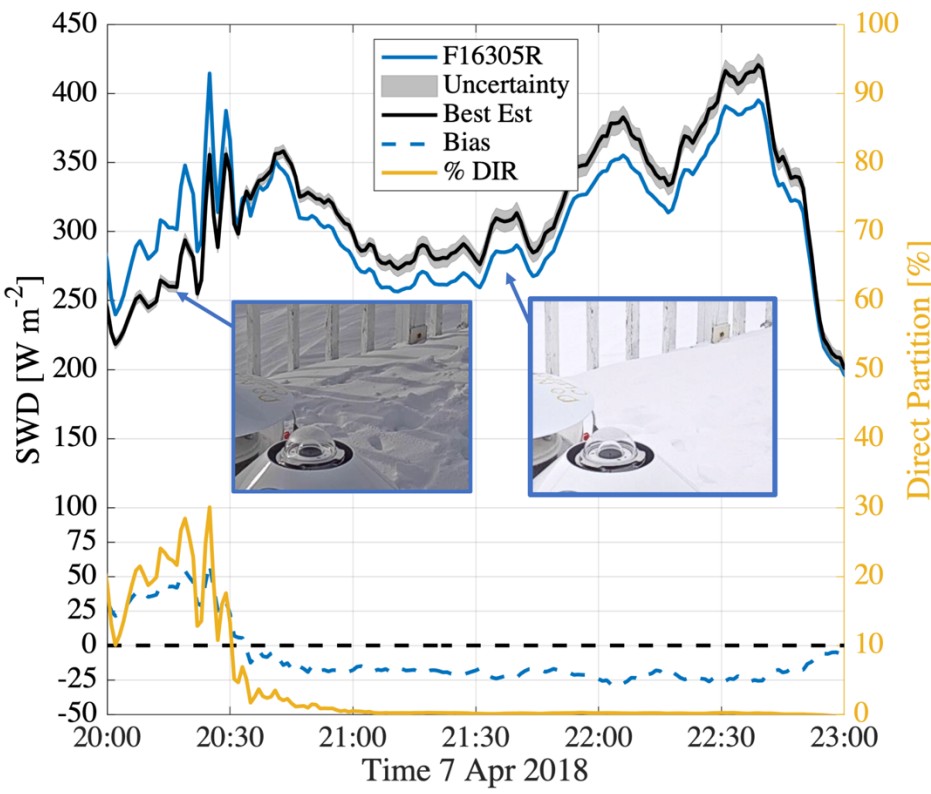

***Figure 5.*** *Case study of SWD on 7 April 2018. The solid black line and gray shading are the best estimate and uncertainty, the solid blue line is an iced pyranometer shown in the inset images. The dashed blue line is the bias in the solid blue relative*

*to the solid black and the yellow line shows the percent of the irradiance contributed by the direct beam: when this value is near zero, the lighting is diffuse under overcast conditions.*






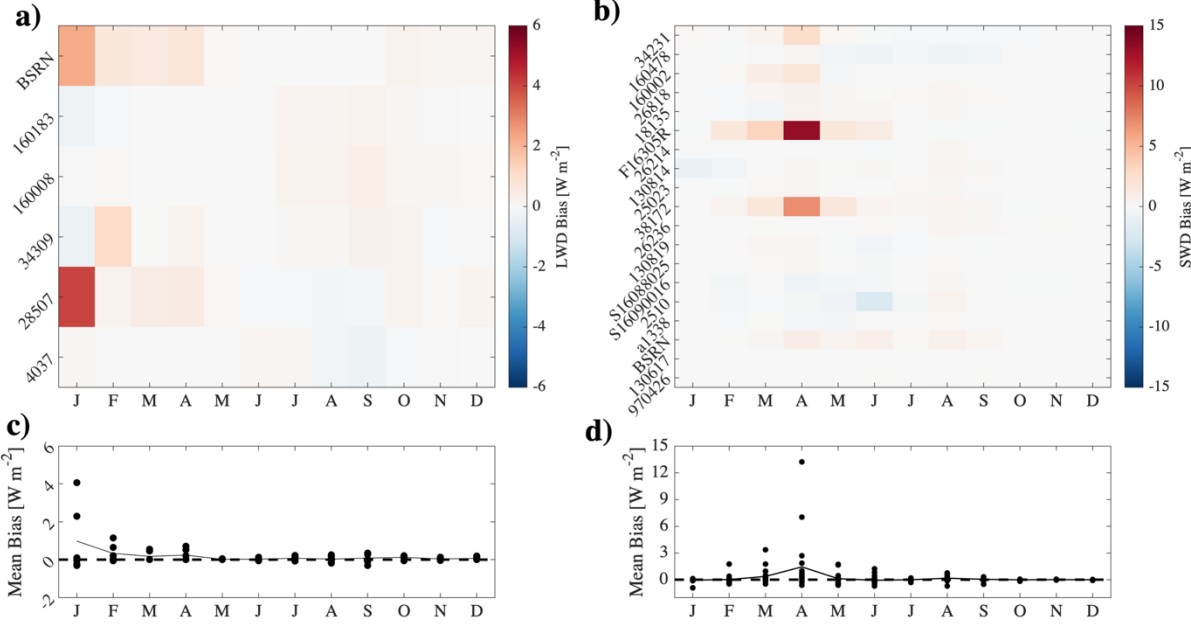

***Figure 6.*** *Calculations of biases in monthly means relative to the BE for SWD (a,c) and LWD (b,d). The upper panels show individual instruments and months; the same data are plotted in time series form in the lower panels for reference. All* 
*calculations are corrected for statistical differences between individual radiometers and the BE when ice-free. The calculations are for all conditions, not just times when ice was present.*



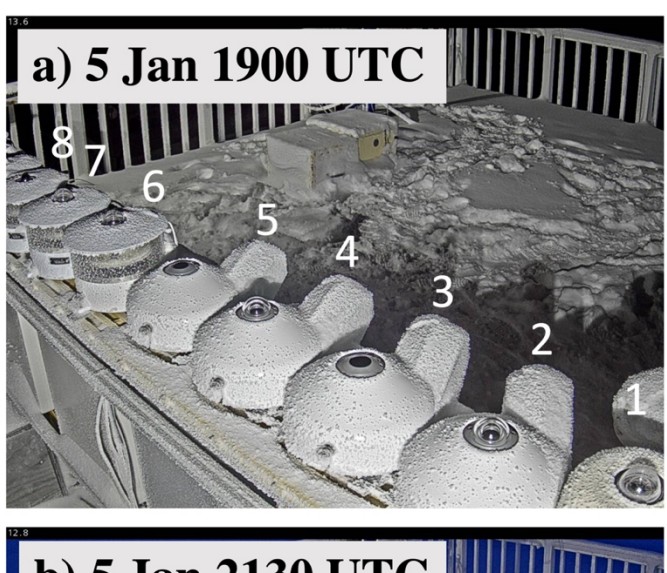

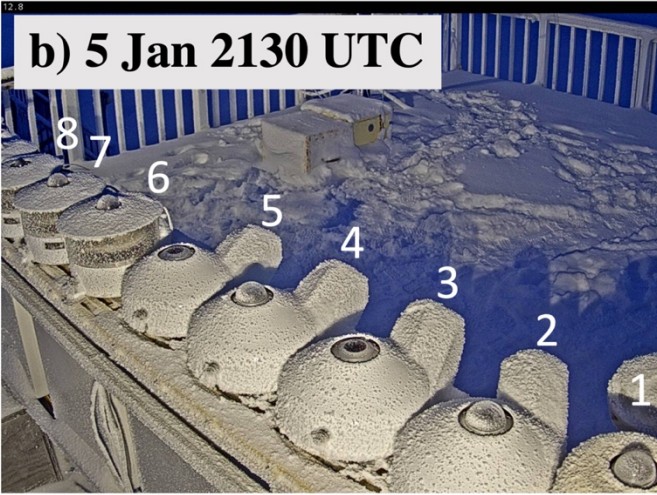


**Figure 7.** *From CAM1, a freezing fog event in progress on 5 Jan and resulting in riming. (a) Ventilators operating normally during the riming event. (b) After the image in (a) was captured, the ventilators were unpowered resulting in icing on domes after several hours of equilibration. Numbers refer to the system positions from Figure 1b.*





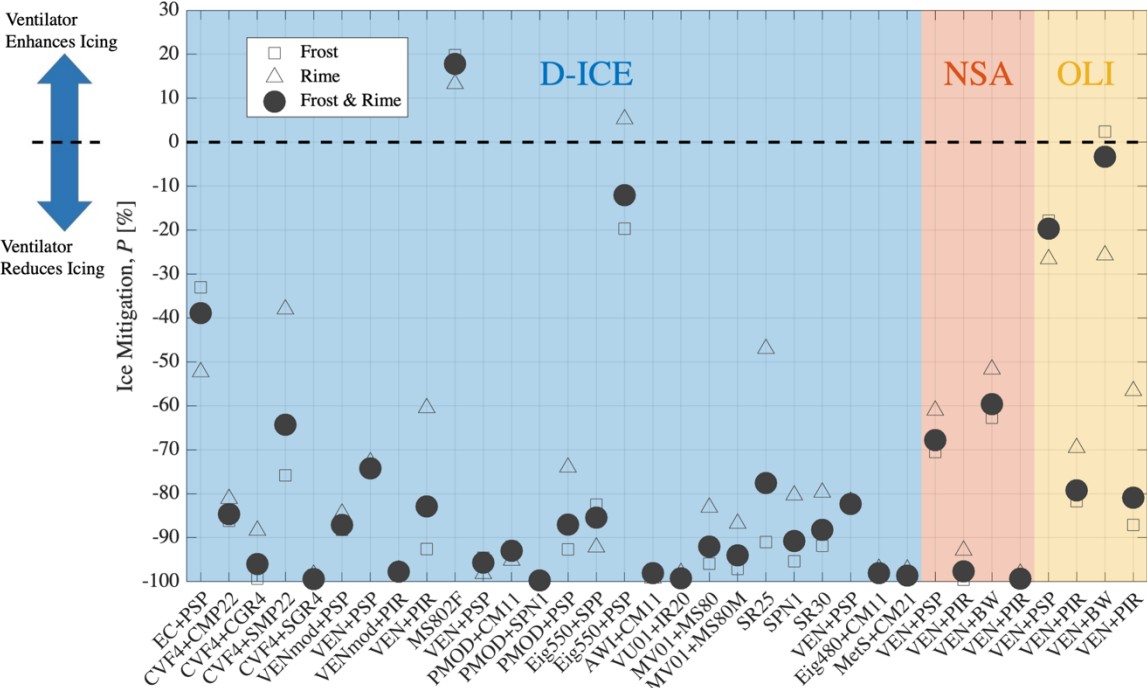


***Figure 8.*** *Ice mitigation performance metric, P, (Eq. 1 from main text) for tested systems at D-ICE (blue background) and DICEXACO (Utqiaġvik "NSA" in red, Oliktok Point "OLI" in yellow). Solid circles refer to a combination of rime and frost while open squares are frost only and open triangles are rime only.*




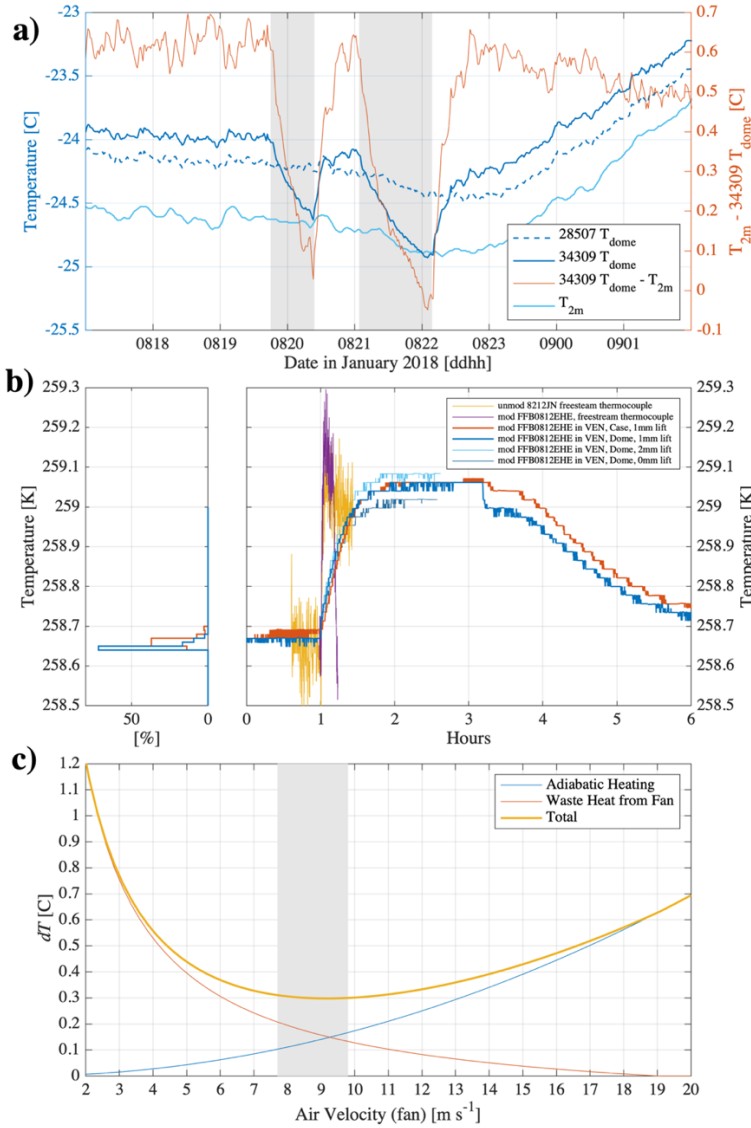

**Figure 9.** (a) Case study from D-ICE on 8-9 Jan, 2018 showing the dome temperatures of PIR 28507 (dashed blue), PIR 34309 (solid blue), the 2 m air temperature (light blue) and the difference in the two dome temperatures (red, right-side axis). The gray shading highlights periods when the ventilator in 34309 was turned off. (b) Laboratory tests carried out in a cold chamber in Boulder, Colorado, using various configurations of the Eppley VEN ventilator housing an FFB0812EHE fan (red and blues) as well as tests with an FFB0812EHE (purple) and 8212JN (yellow) fan without the VEN. The tests are described in Section 4.2. (c) Theoretical calculations of heating from FFB0812EHE fan waste heat (Eq. 2) (red), adiabatic compression (Eq. 3) (blue), and the sum of both (yellow). The gray shading in (c) is the range of observed air velocities near the top of the PIR dome in an Eppley VEN.