# Peer review of "The De-Icing Comparison Experiment (D-ICE): A study of broadband radiometric measurements under icing conditions in the Arctic"

_Atmospheric Measurement Techniques, 2020_

## Referee Comment (RC1) · Anonymous Referee #1 · 17 Nov 2020

This manuscript describes and analyzes results from the De-Icing Comparison Experiment (D-ICE), which aims to provide information to mitigate the long-lasting issue of radiation measurements in polar regions, ice on sensor domes. By comparing 20 pyranometers and 5 pyrgeometers side-by-side during one cold season of the Arctic, the authors present biases in shortwave and longwave radiation measurements caused by icing under different environmental conditions and find that ventilation alone is an effective way to mitigate the issue. They further explain the physical processes of how ventilation reduces icing and provide "the best estimates" (icing free) of radiation measurements for developing quality control procedures to improve retrospective data. The experiment design is appropriate; the findings are novel and practical; the flow and

languages are clear. I recommend publishing on AMT with minor revisions.

Line 102: semi-colon → colon? Line 104: it is not clear if "The operational stations" refer to D-ICE stations or ARM stations or all stations used in this study. Line 193: Did you guys try any machine learning methods to help detect icing situations? Line 226: repeating section numbering Line 291: why the latter data set is "verifiably ice-free"? Line 297: it is not clear whether this "1$\sigma$" is over time or over different instruments Line 309: "When this occurs" meaning > 1 cm or <= 1 cm? Line 319 and 320: panel a → Panel a; panel b → Panel b Line 320: it shows 14-15 April in Fig. 4. Which is it? Line 378: it would be nice to have a mini-summary of shortwave biases here, something similar to Line 344-345 in the longwave section

S1: Since some of the sensors are heated, will this extra heat affect the longwave measurements of surrounding sensors? S1: Does icing ever occur on the camera lenses? How often? How much could it affect your results? S1: the color of model details (shortwave as blue and longwave as red). The number 17 is red but the 17th model name is blue. S2: To confirm, is there any human interference other than in Jan 2018?
* * *

---

## Referee Comment (RC2) · Anonymous Referee #2 · 8 Dec 2020

**Review AMT-2020-397**
**The De-Icing Comparison Experiment (D-ICE): A study of**
**broadband radiometric measurements under icing conditions in the**
**Arctic**

General comments:

The manuscript presents results from "The De-Icing Comparison Experiment" (D-ICE) which was conducted from August 2017 to July 2018 at the NOAA Atmospheric Baseline Observatory in Utqiaġvik, Alaska (71.3°N). In addition, data from the DoE ARM NSA (also Utqiaġvik) and Oliktok Point (250 km east of Utqiaġvik) sites were used. Main objective was to study existing ventilation and heating technologies developed to mitigate radiometer icing. D-ICE comprised 20 pyranometers and 5 pyrgeometers operating in ventilator housings from various vendors alongside operational systems such as BSRN and ARM. Ventilator/heater performance was high with an average ice mitigation at 77% and many being 90 % effective or even better. In addition, a record of ice-free radiative fluxes was compiled. This data set was used to quantify short- and long-term biases in iced sensors. While biases in instantaneous 1-minute records can be up to +60 and -211 to +188 $Wm^{-2}$ for long-wave and shortwave fluxes, respectively, the corresponding biases in the monthly means were substantially smaller at less than approximately 2 $Wm^{-2}$ except for some few systems with insufficient ice mitigation. Finally, observed ice mitigation processes were verified in the laboratory: ventilators without heaters were postulated to be effective by providing heat through waste energy from the fan and adiabatic heating.

The Earth's radiation fluxes play a fundamental role in the climate system, thus accurate ground based observations of shortwave and longwave radiative fluxes are essential for long term monitoring and validation of corresponding satellite products and climate model outputs. Therefore, such experiments are of great importance. D-ICE is an impressive and comprehensive experiment, conducted and documented very carefully. It is unique with respect to its extent providing very useful and novel results.

The manuscript is very well structured and clearly written. The literature has been selected and cited carefully. Graphics and tables are clear and the captions self-explanatory. This work is a very interesting and a valuable contribution to the atmospheric science community and is in my opinion absolutely suited for publication in AMT. I recommend publishing with minor revisions and/or technical corrections.

Specific comments:

- The heater in the 480 unit from Eigenbrodt is operated at 25 W. I expect a substantial warming of the radiometer's body as shown in Fig.1a (on the next page). While the long-wave irradiance is interestingly not necessarily affected (Fig. 1b), the overheating reduces the stability of the pyranometer measurements considerably (see Fig.2). Thus, a high power heating system may boosts ice mitigation but it can affect the observations negatively. Did you also observe higher case temperatures of systems operated at 15 W or higher with respect to heaters running at 10 W?

Some technical corrections:

- Line 310: add 'with respect to windowless long-wave radiometers'
- Line 320: SWD on 13-14 April (but the x-axis of the figure is labelled as 14-15 April)
- Line 383: aggregate means → mean of all sensors? BE product?
- Line 459: Another reasonable explanation would be a different fan speed. However, fan speed was apparently monitored at D-ICE as pointed out later in line 542. You may place this statement already here.
- Line 898 (Caption of Fig. 6): Panel a, c represent LWD (not SWD) and panel b, d SWD (not LWD). Even though the dots in panels c, d may be inferred intuitively from the respective panels a, b, a legend in panels c, d might be helpful to identify the sensors with higher biases highlighted in panels a, b.

[Figure]

*Fig.1: Differences in body temperature (a) and irradiance (b) between two pyrgeometers operated in Eigenbr. 480 ventilation housings. While the heater of the pyrgeometer A1 was continuously run at 10 W, the power of the heater of A2 was increased from 10 to 25 W by end of August.*

[Figure]

*Figure 2: Night data of a pyranometer operated in an Eigenbr. 480 housing. The power of the heater was increased from 10 to 25 W by end of August.*

---

## Referee Comment (RC3) · Anonymous Referee #3 · 12 Dec 2020

This is a review of Cox et al., "The De-Icing Comparison Experiment (D-ICE): A study of broadband radiometric measurements under icing conditions in the Arctic." The authors present a study of the effects of the accumulation of various types of ice (snow, frost or rime) on radiometer domes and of the performance of distinct mitigation systems designed to remove or prevent ice accumulation via ventilation and/or heating. The topic is appropriate AMT.

The study is well-organized and the presentation is clear and well-written. The results are likely to be of significant value to the radiometric measurement community. They will be of use in shaping the selection and design of instruments for future field projects

and in quantifying uncertainties in measurements from instruments currently in use. I am suggesting only a few minor revisions to help with clarity.

Lines 26-27: It's not clear what "mitigating 77%" and "90+% effective" mean, given the limited details available in the abstract. Could these be expressed more concretely here?

Lines 34-36: Could you clarify here whether this is for both shortwave and longwave fluxes?

Line 100: Is there a need here to explain briefly what is meant by "global" downwelling shortwave flux?

Line 170-171: What is meant by "rime or frost was observed to be present in the vicinity of the D-ICE systems"? This and the associated paragraph are a bit unclear. In particular, what is meant by "sublimation period" and by "duration of the presence of ice"? Does this mean ice on the radiometer domes, or ice evident elsewhere?

Lines 285-286: Note that in ARM parlance, "NSA" encompasses both BRW and OLI. Are the included NSA radiometers at BRW?

Line 300: No relationship was observed between "the number of radiometers included in estimated uncertainty" and what?

Line 352: This goes back to my question about "sublimation period" for Lines 170-171. As far as I can determine, this is the first use of the term "deposition period." It would be helpful to clearly define this and "sublimation period" earlier in the paper.

Line 422 and elsewhere: Ditto my earlier comment re. "NSA" vs. "BRW" vs. "OLI".

Line 425: So is t_icing the same as the length of the "deposition period"? Maybe try to standardize a bit (e.g., "icing period" instead of "deposition period").

Line 476: "When outside of the ventilator", this means when the *fan* is operated outside of a ventilator?

Lines 499-508: It would be interesting to see the results of a similar experiment but with the flow direction of the fan reversed. There would be no fan waste heat warming the air flowing over the dome and no adiabatic compression effects near the dome.

Line 529: Should this be "specific volume \*at\* the total pressure"?

Line 560-561: Maybe be more specific than "amount of ice", since this could be confused with mass of deposited ice.

---

## Author Comment (AC1) · 28 Dec 2020

Thank you to the Reviewer for providing constructive and thoughtful feedback, which have helped us to improve the manuscript. Our point-by-point responses are provided below in blue text following the Reviewer's comments, reproduced in black.

**Review 1 Comments (RC1)**

This manuscript describes and analyzes results from the De-Icing Comparison Experiment (D-ICE), which aims to provide information to mitigate the long-lasting issue of radiation measurements in polar regions, ice on sensor domes. By comparing 20 pyranometers and 5 pyrgeometers side-by-side during one cold season of the Arctic, the authors present biases in shortwave and longwave radiation measurements caused by icing under different environmental conditions and find that ventilation alone is an effective way to mitigate the issue. They further explain the physical processes of how ventilation reduces icing and provide "the best estimates" (icing free) of radiation measurements for developing quality control procedures to improve retrospective data. The experiment design is appropriate; the findings are novel and practical; the flow and languages are clear. I recommend publishing on AMT with minor revisions.

Line 102: semi-colon → colon?
We replaced the semi-colon with a comma and split the statement into two sentences.

Line 104: it is not clear if "The operational stations" refer to D-ICE stations or ARM stations or all stations used in this study.
We refer to the radiometric stations at NOAA-GML (BSRN), ARM-NSA, and ARM-OLI, which are similarly configured. We have clarified this in the text.

Line 193: Did you guys try any machine learning methods to help detect icing situations?
We considered machine learning and prior to carrying out the manual classifications we discussed the possibility with a machine learning expert at a DoE Atmospheric Systems Research (ASR) conference. In the end we decided not to attempt to implement it out of concern for a large number of potential false positives and negatives due to irregularity in the shape and texture of ice in the images. Additionally, the size of the training set would have likely been similar in size to the base data set of images.

Line 226: repeating section numbering
Thank you for identifying this error. We have corrected it.

Line 291: why the latter data set is "verifiably ice-free"?
These were the data that were monitored directly using the cameras and were visually verified as ice-free as described in Section 2.3.1. We have clarified the text.

Line 297: it is not clear whether this "1σ" is over time or over different instruments
It is the latter. We have clarified this in the text.

Line 309: "When this occurs" meaning > 1 cm or <= 1 cm?
It is the latter. We have clarified this in the text.

Line 319 and 320: panel a → Panel a; panel b → Panel b
Based on AMT's author guide, we think the change actually needs to be panel a -> panel (a); panel b -> panel (b). We have made this change, as well as similar changes elsewhere in the text.

Line 320: it shows 14-15 April in Fig. 4. Which is it?
Thank you for catching this error. It is 14-15. We have corrected the text.

Line 378: it would be nice to have a mini-summary of shortwave biases here, something similar to Line 344-345 in the longwave section
We have added the following sentence at the end of the section: "These cases demonstrate that errors from ice in SWD can be large and that the sign of the bias is dependent on the amount of coverage of ice on the pyranometer dome, as well as the presence, and likely also the angle, of irradiance from the direct beam."

S1: Since some of the sensors are heated, will this extra heat affect the longwave measurements of surrounding sensors?
We concede that we cannot rule out this possibility. However, as described in section 2.1, the orientation and layout of the instruments was designed so that each instrument was exposed to the predominant winds. We expanded this discussion including specific reference to the wind direction, citing an earlier study we conducted that reported on those winds.

S1: Does icing ever occur on the camera lenses? How often? How much could it affect your results?
It can, but rarely. We designed the camera installation to minimize this problem and were successful. In section 2.1 we state that "All systems on the D-ICE table were monitored using three 720p low-light (0.1 lux) cameras in heated enclosures. … They were installed facing west (away from the predominant wind direction). … The cameras were functional and unobscured by ice for 97.6% of the campaign." Indeed, only a portion of the small amount of downtime is attributable to ice, while the rest was due to a power outage.

S1: the color of model details (shortwave as blue and longwave as red). The number 17 is red but the 17th model name is blue.
Thank you for point this out. There were actually two other similar color-coding errors. We made the corrections.

S2: To confirm, is there any human interference other than in Jan 2018?
We reviewed our notes and found several additional instances of minor modifications, which we have added to S2:
October 13: added putty to radiometer 6 and 8
October 26: removed intake screens from 2-5, 15-17, 12-14
February 8: repaired fan at #14 20:54Z
March 20: instrument levels checked and corrected
We have also explicitly noted (in text) the relevant dates when ice was cleaned from radiometers that was discussed generally in Section 2.1.

---

## Author Comment (AC2) · 28 Dec 2020

Thank you to the Reviewer for providing constructive and thoughtful feedback, which have helped us to improve the manuscript. Our point-by-point responses are provided below in blue text following each the Reviewer's comments, reproduced in black.

**Review 2 Comments (RC2)**

General comments:

The manuscript presents results from "The De-Icing Comparison Experiment" (D-ICE) which was conducted from August 2017 to July 2018 at the NOAA Atmospheric Baseline Observatory in Utqiaġvik, Alaska (71.3°N). In addition, data from the DoE ARM NSA (also Utqiaġvik) and Oliktok Point (250 km east of Utqiaġvik) sites were used. Main objective was to study existing ventilation and heating technologies developed to mitigate radiometer icing. D-ICE comprised 20 pyranometers and 5 pyrgeometers operating in ventilator housings from various vendors alongside operational systems such as BSRN and ARM. Ventilator/heater performance was high with an average ice mitigation at 77% and many being 90 % effective or even better. In addition, a record of ice- free radiative fluxes was compiled. This data set was used to quantify short- and long-term biases in iced sensors. While biases in instantaneous 1-minute records can be up to +60 and -211 to +188 Wm-2 for long-wave and shortwave fluxes, respectively, the corresponding biases in the monthly means were substantially smaller at less than approximately 2 Wm-2 except for some few systems with insufficient ice mitigation. Finally, observed ice mitigation processes were verified in the laboratory: ventilators without heaters were postulated to be effective by providing heat through waste energy from the fan and adiabatic heating.

The Earth's radiation fluxes play a fundamental role in the climate system, thus accurate ground based observations of shortwave and longwave radiative fluxes are essential for long term monitoring and validation of corresponding satellite products and climate model outputs. Therefore, such experiments are of great importance. D-ICE is an impressive and comprehensive experiment, conducted and documented very carefully. It is unique with respect to its extent providing very useful and novel results.

The manuscript is very well structured and clearly written. The literature has been selected and cited carefully. Graphics and tables are clear and the captions self-explanatory. This work is a very interesting and a valuable contribution to the atmospheric science community and is in my opinion absolutely suited for publication in AMT. I recommend publishing with minor revisions and/or technical corrections.

Specific comments:

The heater in the 480 unit from Eigenbrodt is operated at 25 W. I expect a substantial warming of the radiometer's body as shown in Fig.1a (on the next page). While the long-wave irradiance is interestingly not necessarily affected (Fig. 1b), the overheating reduces the stability of the pyranometer measurements considerably (see Fig.2). Thus, a high power heating system may boosts ice mitigation but it can affect the observations negatively. Did you also observe higher case temperatures of systems operated at 15 W or higher with respect to heaters running at 10 W?

Your question is important and we recognize this issue. It was in fact one of our motivations, though we approached the problem differently beginning with the hypothesis put forward by the BSRN working group that "aspiration of ambient air without additional heat is sufficient to mitigate ice". Because the experimental design was built around explaining how this could be achieved, the D-ICE data have limited potential for studying the negative effects of heating. The campaign did not feature enough configurations of heaters using the same models of ventilator and radiometer, and most systems housed pyranometers, many of which do not have case thermistors. As it were, the nighttime offsets (and so the signal we would be looking for) observed during D-ICE were small, in part due to the models of pyranometers, as well as environmental conditions (like frequent cloud cover) and happenstance. We concluded that "…while heating elements were found to be effective, they are not required for successful ice mitigation" alongside details explaining this observation.

Some technical corrections:
- Line 310: add 'with respect to windowless long-wave radiometers'
We have added this phrase to the text.

- Line 320: SWD on 13-14 April (but the x-axis of the figure is labelled as 14-15 April)
Thank you for pointing this out. The figure is correct. We have changed the text.

- Line 383: aggregate means◊mean of all sensors? BE product?
Yes, it is the composite mean of the mean errors from the individual sensors. The calculation is relative to the BE product. We have added clarity to the text.

- Line 459: Another reasonable explanation would be a different fan speed. However, fan speed was apparently monitored at D-ICE as pointed out later in line 542. You may place this statement already here.
The statements at line 542 are specific to the Eppley VEN experiments, but we agree that they apply also here. Based on your suggestion, we reviewed the fan speed data from the ventilators discussed at line 459 and we found that the fan speeds agreed with one another within about 1% for the duration of the experiment, though interestingly they were about 7% lower than the same fans in KZ ventilators housing pyrgeometers for most of the winter into the spring. Your suggestion was a good one but does not appear to be the explanation so we have not made changes to the text. Notably, the conclusions from section 4.2 (Fig 9c) indicate that small differences in fan speed – assuming these differences are expressions of variability in fan efficiency – actually do not necessarily cause much net difference in dome heating, but rather exchange the source of the heating between adiabatic heating and waste heat.

- Line 898 (Caption of Fig. 6): Panel a, c represent LWD (not SWD) and panel b, d SWD (not LWD). Even though the dots in panels c, d may be inferred intuitively from the respective panels a, b, a legend in panels c, d might be helpful to identify the sensors with higher biases highlighted in panels a, b.
Thank you for identifying this error, which we have corrected. After consideration of your suggestion for additional labeling, we have decided that we prefer it as is because we cannot

think of a way to do this without making the graphic very busy and, as you say, it would not add any new information to the figure.

---

## Author Comment (AC3) · 28 Dec 2020

Thank you to the Reviewer for providing constructive and thoughtful feedback, which have helped us to improve the manuscript. Our point-by-point responses are provided below in blue text following the Reviewer's comments, reproduced in black.

**Review 3 Comments (RC3)**

This is a review of Cox et al., "The De-Icing Comparison Experiment (D-ICE): A study of broadband radiometric measurements under icing conditions in the Arctic." The authors present a study of the effects of the accumulation of various types of ice (snow, frost or rime) on radiometer domes and of the performance of distinct mitigation systems designed to remove or prevent ice accumulation via ventilation and/or heating. The topic is appropriate AMT.

The study is well-organized and the presentation is clear and well-written. The results are likely to be of significant value to the radiometric measurement community. They will be of use in shaping the selection and design of instruments for future field projects and in quantifying uncertainties in measurements from instruments currently in use. I am suggesting only a few minor revisions to help with clarity.

Lines 26-27: It's not clear what "mitigating 77%" and "90+% effective" mean, given the limited details available in the abstract. Could these be expressed more concretely here?
We have clarified the statement, which now reads as follows: "Ventilator and ventilator/heater performance overall was skilful with the average of the systems mitigating ice formation 77% (many > 90%) of the time during which icing conditions were present."

Lines 34-36: Could you clarify here whether this is for both shortwave and longwave fluxes?
Yes, we added "in both the shortwave and longwave."

Line 100: Is there a need here to explain briefly what is meant by "global" downwelling shortwave flux?
Yes, we have clarified the jargon.

Line 170-171: What is meant by "rime or frost was observed to be present in the vicinity of the D-ICE systems"? This and the associated paragraph are a bit unclear. In particular, what is meant by "sublimation period" and by "duration of the presence of ice"? Does this mean ice on the radiometer domes, or ice evident elsewhere?
In Section 2.2 we are characterizing the natural icing events that were observed and so we are referring to ice appearing on surfaces other than the domes. We have added an introductory sentence at the beginning of the section to clarify. We have also clarified this again in the first sentence of the section's second paragraph. When ice is present, it may be developing (e.g., deposition) or disappearing (sublimation since the temperatures were almost always below freezing). We referred to the latter condition as a "sublimation period", but there really is no reason to define such a term since it is only used once. Therefore, we have replaced the term with a descriptive statement, "portion of time the ice was present and sublimating". We have also added clarity to the statement of duration as "… duration of presence of ice on surfaces surrounding the experiment."

Lines 285-286: Note that in ARM parlance, "NSA" encompasses both BRW and OLI. Are the included NSA radiometers at BRW?

You are correct that "north slope Alaska" is a definition that encompasses the broader area. Unfortunately, the term "NSA" as used by the ARM community has become ambiguous during the development of the ARM observatories at Barrow, Atqasuk, and Oliktok. Atqasuk and Barrow for example are designated the C1 and C2 facilities of NSA (with E* subsites), but the addition of Oliktok in 2013 did not create a C3 NSA facility. Rather, ARM considers Oliktok to be a separate deployment, in part because of the fact that it is a mobile facility (AMF-3) and not a permanent installation. We have spoken with the site scientist for Oliktok, who has confirmed that while NSA refers to Barrow and surrounding, OLI is separate. Our choice of NSA and OLI as reference identifiers is consistent with the file naming conventions used in ARM's data streams that we analyze (Atqasuk = atq and Barrow = nsa), which we believe is convenient for readers or those who want to have a look at the data themselves. We agree that the conventions can be confusing but we were explicit in the geographical descriptions at the top of Section 2.1 where we define our use of initialisms. We would prefer to keep these conventions.

Line 300: No relationship was observed between "the number of radiometers included in estimated uncertainty" and what?

A limitation of the uncertainty calculation is that removal of data during quality control screening means that at some times fewer measurements are available for the calculation than at other times. We were concerned that this could affect the robustness of the metric. We found that the calculated uncertainty is uncorrelated with the number of values that were used to calculate it, which is evidence that our concern was not significantly impactful. We have rewritten the statement in a way that we hope is clearer.

Line 352: This goes back to my question about "sublimation period" for Lines 170-171. As far as I can determine, this is the first use of the term "deposition period." It would be helpful to clearly define this and "sublimation period" earlier in the paper.

As before, there is no reason for us to define such a term since it is only used once so we have replaced it with a descriptive statement, "…time during which the frost was observed to be growing through deposition."

Line 422 and elsewhere: Ditto my earlier comment re. "NSA" vs. "BRW" vs. "OLI".

Please refer to our response to the earlier comment.

Line 425: So is t_icing the same as the length of the "deposition period"? Maybe try to standardize a bit (e.g., "icing period" instead of "deposition period").

No, t_icing is the "amount of time icing conditions occurred", which would be the total length of time of deposition followed by sublimation. The distinction is not important for t_icing and since the confusing terms "sublimation/deposition period" have already been removed from the text, we have not made any additional changes with respect to this comment.

Line 476: "When outside of the ventilator", this means when the *fan* is operated outside of a ventilator?

Correct. We have clarified the statement.

Lines 499-508: It would be interesting to see the results of a similar experiment but with the flow direction of the fan reversed. There would be no fan waste heat warming the air flowing over the dome and no adiabatic compression effects near the dome.

This is an excellent suggestion! We would like to try this right now if it weren't for the fact that COVID-19 protocols prevent us from accessing the lab. In thinking about it, we suspect there might actually be adiabatic processes occurring within the ventilator chamber because the air pressure within the chamber would likely be lowered by the action of the fan. This would cause some cooling that could affect the case. It is less clear what would happen at the dome because the air flow around the dome with air being drawn in instead of forced out of the vent would probably follow a different path.

Line 529: Should this be "specific volume *at* the total pressure"?

Yes, something is amiss. It should be "…specific volume *of the air at* total pressure…". We have made this change.

Line 560-561: Maybe be more specific than "amount of ice", since this could be confused with mass of deposited ice.

Yes, this was ambiguous. We have added clarity to the text.